# Taste triggers a homeostatic temperature control in hungry flies

**Yujiro Umezaki[1], Sergio Hidalgo[2], Erika Nguyen[3], Tiffany Nguyen[3], Jay Suh[3], Sheena S Uchino[1], Joanna Chiu[2], Fumika Hamada[1]***

[1]Department of Neurobiology, Physiology and Behavior, University of California, Davis, Davis, United States; [2]Department of Entomology and Nematology, University of California, Davis, Davis, United States; [3]Division of Developmental Biology, Cincinnati Children's Hospital Medical Center, Cincinnati, United States

## eLife assessment

This paper presents **valuable** findings that gustation and nutrition might independently influence the preferred environmental temperature in flies. The evidence supporting the main claims is **solid** and well presented. The finding that flies might thus exhibit a cephalic phase response similar to mammals will be of value for future investigations.

**\*For correspondence:**
fnhamada@ucdavis.edu

**Abstract** Hungry animals consistently show a desire to obtain food. Even a brief sensory detection of food can trigger bursts of physiological and behavioral changes. However, the underlying mechanisms by which the sensation of food triggers the acute behavioral response remain elusive. We have previously shown in *Drosophila* that hunger drives a preference for low temperature. Because *Drosophila* is a small ectotherm, a preference for low temperature implies a low body temperature and a low metabolic rate. Here, we show that taste-sensing triggers a switch from a low to a high temperature preference in hungry flies. We show that taste stimulation by artificial sweeteners or optogenetics triggers an acute warm preference, but is not sufficient to reach the fed state. Instead, nutrient intake is required to reach the fed state. The data suggest that starvation recovery is controlled by two components: taste-evoked and nutrient-induced warm preferences, and that taste and nutrient quality play distinct roles in starvation recovery. Animals are motivated to eat based on time of day or hunger. We found that clock genes and hunger signals profoundly control the taste-evoked warm preferences. Thus, our data suggest that the taste-evoked response is one of the critical layers of regulatory mechanisms representing internal energy homeostasis and metabolism.

## Introduction

Animals are constantly sensing environmental stimuli and changing their behavior or physiology based on their internal state (*Root et al., 2011*; *Kain and Dahanukar, 2015*; *Soria-Gómez et al., 2014*; *Hanci and Altun, 2016*; *Keys et al., 1950*; *Piccione et al., 2002*; *Sakurada et al., 2000*). Hungry animals are strongly attracted to food. Immediately after seeing, smelling, or chewing food, even without absorbing nutrients, a burst of physiological changes is suddenly initiated in the body. These responses are known in mammals as the cephalic phase response (CPR) (*Smeets et al., 2010*). For example, a flood of saliva and gastrointestinal secretions prepares hungry animals to digest food (*Chen and Knight, 2016*; *Power and Schulkin, 2008*; *Zafra et al., 2006*; *Pavlov, 1902*; *Garrison and Knight, 2017*). Starvation results in lower body temperatures, and chewing food triggers a rapid increase in heat production, demonstrating CPR in thermogenesis (*LeBlanc, 2000*; *LeBlanc and*

*Cabanac, 1989*; *LeBlanc et al., 1984*). However, the underlying mechanisms of how the sensation of food without nutrients triggers the behavioral response remain unclear.

To address this question, we used a relatively simple and versatile model organism, *Drosophila melanogaster*. Flies exhibit robust temperature preference behavior (*Sayeed and Benzer, 1996*; *Dillon et al., 2009*). Due to the low mass of small ectotherms, the source of temperature comes from the environment. Therefore, their body temperatures are close to the ambient temperature (*Stevenson, 1985a*; *Stevenson, 1985b*). For temperature regulation, animals are not simply passive receivers of ambient temperature. Instead, they actively choose an ambient temperature based on their internal state. For example, we have shown that preferred temperature increases during the daytime and decreases during the night time, exhibiting circadian rhythms of temperature preference (temperature preference rhythms: TPR) (*Kaneko et al., 2012*). Because their surrounding temperature is very close to their body temperature, TPR leads to body temperature rhythms (BTR) that is very similar to mammalian BTR (*Goda and Hamada, 2019*; *Goda et al., 2023*). Another example is the starvation. Hunger stress forces flies to change their behavior and physiological response (*Lin et al., 2019*; *Zhang et al., 2019*). We previously showed that the hungry flies prefer a lower temperature (*Umezaki et al., 2018*). The flies in a lower environmental temperature has been shown to have a lower metabolic rate, and the flies in a higher environmental temperature has been shown to have a higher metabolic rate (*Umezaki et al., 2018*; *Berrigan and Partridge, 1997*; *Schilman et al., 2011*). Therefore, hungry flies choose a lower temperature and therefore, their metabolic rate is lower. Similarly, in mammals, starvation causes a lower body temperature, hypothermia (*Piccione et al., 2002*). In mammals, body temperature is controlled by the balance between heat loss and heat production. The starved mammals have been shown a lower heat production (*Keys et al., 1950*; *Piccione et al., 2002*; *Sakurada et al., 2000*). Therefore, both flies and mammals, the starvation causes a low body temperature.

The flies exhibit robust feeding behaviors (*Ja et al., 2007*; *Itskov et al., 2014*) and molecular and neural mechanisms of taste are well documented (*Kain and Dahanukar, 2015*; *Wang and Wang, 2019*; *Pool and Scott, 2014*; *Snell et al., 2022*; *Kim et al., 2017*; *Liu et al., 2015*; *Frank et al., 2015*). Therefore, we focused on taste and temperature regulation and asked how the taste cue triggers a robust behavioral recovery of temperature preference in starving flies. We show in hungry flies that taste without nutrients induces a switch from a low to a high temperature preference. While taste leads to a warmer temperature preference, nutrient intake causes the flies to prefer an even warmer temperature. This nutrient-induced warm preference results in a complete recovery from starvation. Thus, taste-evoked warm preference is different from nutrient-induced warm preference and potentially similar physiology as CPR. Therefore, when animals emerge from starvation, they use a two-step approach to recovery: taste-evoked and nutrient-induced warm preference. While a rapid component is elicited by food taste alone, a slower component requires nutrient intake.

Animals are motivated to eat based on their internal state, such as time of day or degree of hunger. The circadian clock drives daily feeding rhythms (*Cedernaes et al., 2019*; *Longo and Panda, 2016*; *Hirayama et al., 2018*) and anticipates meal timing. Daily feeding timing influences energy homeostasis and metabolism (*Greenhill, 2018*; *Chaix et al., 2019*). Circadian clocks control feeding behavior in part via orexigenic peptidergic/hormonal regulation such as neuropeptide Y (NPY) and agouti-related peptide (AgRP) neurons, which are critical for regulating feeding and metabolism (*Hirayama et al., 2018*; *Blasiak et al., 2017*). Feeding–fasting cycles modulate peripheral organs in liver, gut, pancreas, and so on (*Panda, 2016*), suggesting that circadian clocks, peptidergic/hormonal signals, and peripheral organs are organically coordinated to enable animals maintain their internal states constantly. We found that clock genes and hunger signals are strongly required for taste-evoked warm preference. The data suggest that taste-evoked response is an indispensable physiological response that represents internal state. Taken together, our data shed new light on the role of the taste-evoked response and highlight a crucial aspect of our understanding of feeding state and energy homeostasis.

## Impact statement

Hungry flies shifting from cooler to warmer temperature preferences in response to non-nutritive food indicates a taste-triggered response, similar to the CPR observed in mammals.

## Results

### Food detection triggers a warm preference

To investigate how food detection influences temperature preference behavior in *Drosophila* (*Sayeed and Benzer, 1996*; *Dillon et al., 2009*), the *white*$^{1118}$ (*w*$^{1118}$) control flies were fed fly food containing carbohydrate, protein, and fat sources and tested in temperature preference behavioral assays (*Figure 1A–C*; *Umezaki et al., 2018*; *Hamada et al., 2008*). The flies are released into a chamber set to a temperature range of 16–34°C and subsequently accumulate at their preferred temperature (Tp) at 25.2 ± 0.2°C within 30 min (*Figure 1D*: fed, white bar). On the other hand, when *w*$^{1118}$ flies were starved overnight with water only, they preferred 21.7 ± 0.3°C (*Figure 1D*: overnight starvation (STV), gray bar). Thus, starvation leads to a lower Tp. As we have previously reported, starvation strongly influences temperature preference (*Umezaki et al., 2018*).

To examine how starved flies recover from lower Tp, they were offered fly food for 5 min, 10 min, 30 min, and 1 hr. Immediately after the flies were refed, the temperature preference behavior assay was examined. The assay takes 30 min from the time the flies are placed in the apparatus until the final choice is made. For example, in the case of 5 min refed flies, it took a total of 35 min from the start of refeeding to the end of the assay. After 10 min, 30 min, or 1 hr of fly food refeeding, starved flies preferred a temperature similar to that of fed flies (*Figure 1D*, *Figure 1—figure supplement 1*, and *Figure 2—figure supplement 1*: orange bar, statistics shown as red stars, *Figure 1— source data 1*), suggesting a full recovery from starvation. On the other hand, refeeding after 5 min resulted in a warmer temperature than the starved flies. Nevertheless, Tp did not reach that of the fed flies (*Figure 1D*, *Figure 1—figure supplement 1*, and *Figure 2—figure supplement 1*). Therefore, refeeding the flies for 5 min resulted in a partial recovery from the starved state (*Figure 1*, *Figure 1— figure supplement 1*, and *Figure 2—figure supplement 1*: statistics shown as green stars, *Figure 1—source data 1*). Thus, our data suggest that food intake triggers a warm preference in starved flies.

### Sucralose refeeding promotes a warm preference

While only 5 min of refeeding fly food caused hungry flies to prefer a slightly warmer temperature, 10 min of refeeding caused hungry flies prefer a similarly warmer temperature as the fed flies (*Figure 1D*, *Figure 1—figure supplement 1*, and *Figure 2—figure supplement 1*). Therefore, we hypothesized that food-sensing cues might be important for the warm preference. Sucralose is an artificial sweetener that activates sweet taste neurons in *Drosophila* (*Biolchini et al., 2017*) and modulates taste behaviors such as the proboscis extension reflex (*Park et al., 2017*; *Dahanukar et al., 2007*; *Dus et al., 2011*; *Wang et al., 2016*). Importantly, sucralose is a non-metabolizable sugar and has no calories. Therefore, to investigate how food-sensing cues are involved in warm preference, we examined how sucralose refeeding changes the temperature preference of starved flies. After starved flies were refed sucralose for 10 min or 1 hr, they preferred a warmer temperature; however, Tp was halfway between Tps of fed and starved flies (*Figure 1E*, *Figure 1—figure supplement 1*, and *Figure 2—figure supplement 1*: blue bar, *Figure 1—source data 1*). Thus, sucralose ingestion induces a warm preference but shows a partial recovery of Tp from the starved state.

While refeeding the flies with food resulted in a full recovery of Tp from the starved state, refeeding them with sucralose resulted in only a partial recovery of Tp. Therefore, starved flies may use both taste cues and nutrients to fully recover Tp from the starved state. To evaluate this possibility, we used glucose, which contains both sweetness (gustatory cues) and nutrients (i.e., metabolizable sugars), and tested glucose refeeding for 5 min, 10 min, and 1 hr. We found that 10 min or 1 hr glucose refeeding resulted in full recovery of Tp from the starved state (*Figure 1F*, *Figure 1—figure supplement 1*, and *Figure 2—figure supplement 1*: green bar, statistics shown in red NS, *Figure 1—source data 1*) and was significantly different from starved flies (*Figure 1F*, *Figure 1—figure supplement 1*, and *Figure 2—figure supplement 1*: statistics shown in green stars, *Figure 1—source data 1*). Thus, our data showed that sucralose refeeding induced partial recovery and glucose refeeding induced full recovery from the starved state.

It is still possible that the starved flies consumed glucose faster than sucralose during the first 10 min, which could result in a different warming preference. To rule out this possibility, we examined how often starved flies touched glucose, sucralose, or water during the 30 min using the Fly Liquid-food Interaction Counter (FLIC) system (*Ro et al., 2014*). The FLIC system assays allow us to monitor how much interaction between the fly and the liquid food reflects feeding episodes. We found that

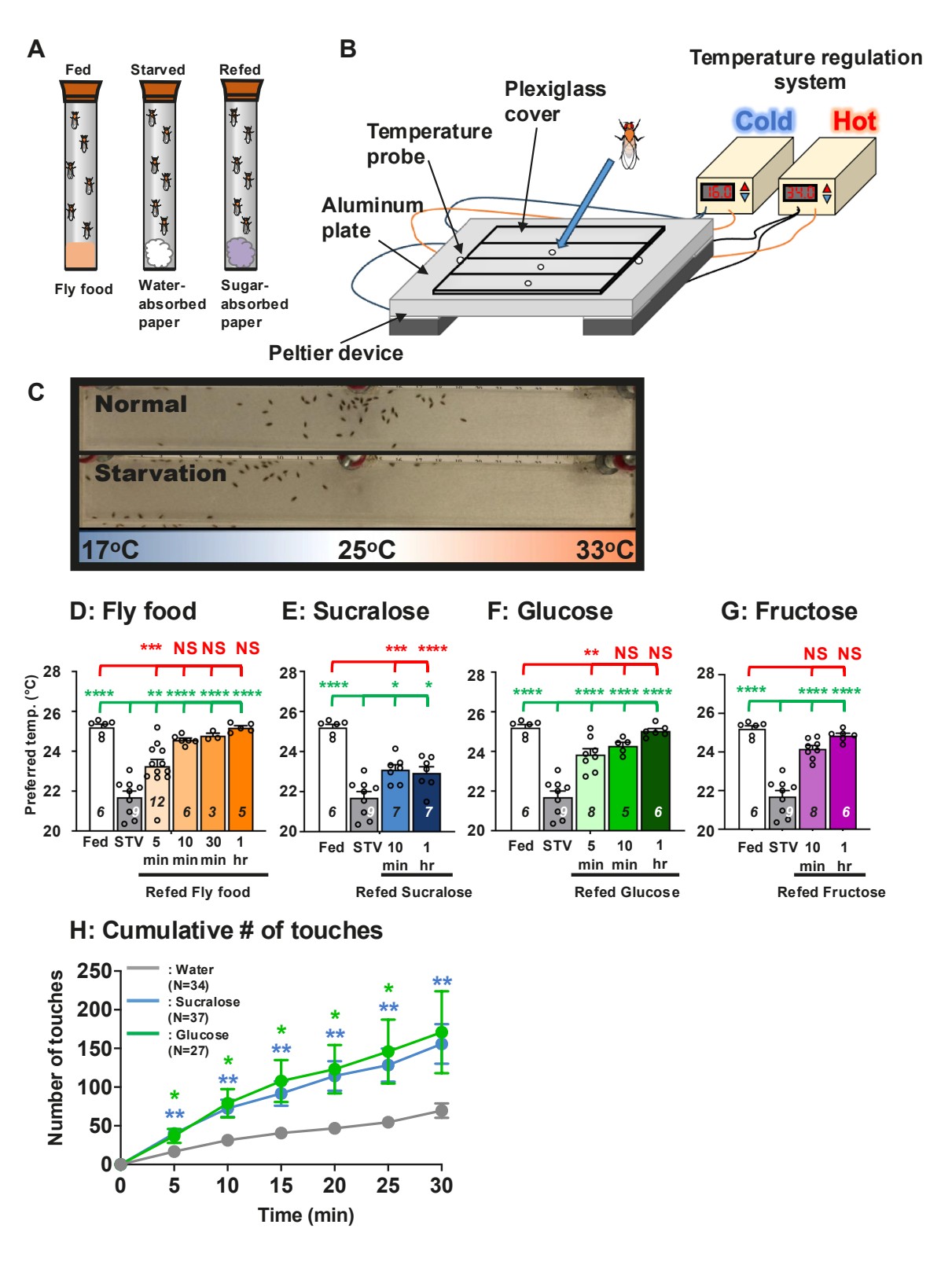

**Figure 1.** Hungry flies switch from cold to warm preference upon food detection. (**A**) Three different feeding conditions, fed, starved and refed, were used in this study. (**B**) A temperature gradient of 17–33°C in air is established in the chamber between the aluminum metal plate and the plexiglas cover. Flies are applied through the hole into the chamber. Once the flies are applied, they spread out in the chamber and then gradually accumulate at the specific temperature ranges within 30 min, which is called the preferred temperature (Tp). (**C**) One of the representative results of Tp experiments (top,

*Figure 1 continued on next page*

*Figure 1 continued*

normal (fed flies); bottom, starvation (starved flies)). Since their body temperature is close to the temperature of their surrounding microenvironment, their body temperature is determined by measuring their Tp. (**D–G**) Comparison of preferred temperature (Tp) of *white*[1118] (*w*[1118]) control flies between fed (white bar), starved (STV; gray bar), and refed (orange, blue, or green bar) states. Starvation was applied for 1 overnight (ON). Starved flies were refed with fly food (orange bar) for 5, 10, 30, or 60 min (1 hr) (**D**), 2.8 mM sucralose solution (blue bar) for 10 min or 1 hr (**E**), 2.8 mM (equivalent to 5%) glucose solution (green bar) for 5 min, 10 min, or 1 hr (**F**), or 2.8 mM fructose solution (purple bar) for 10 min or 1 hr (**G**). Behavioral experiments were performed at the specific time points, Zeitgeber time (ZT) 4–7. ZT0 and ZT12 are light on and light off, respectively. Dots on each bar indicate individual Tp in the assays. Numbers in italics indicate the number of trials. The Shapiro–Wilk test was performed to test for normality. One-way ANOVA was performed to compare Tp between each refeeding condition. Red or green stars indicate Tukey's post hoc test comparing differences between experimental and fed or starved conditions, respectively. Data are presented as mean Tp with SEM. *$p < 0.05$. **$p < 0.01$. ***$p < 0.001$. ****$p < 0.0001$. NS indicates no significance. (**H**) Feeding assay: The number of touches to water (gray), 2.8 mM (equivalent to 5%) glucose (green), or 2.8 mM sucralose in each solution (blue) was examined using *w*[1118] flies starved for 24 hr. Water, glucose, and sucralose were tested individually in the separate experiments. A cumulative number of touches to water or sugar solution for 0–30 min was plotted. Two-way ANOVA was used for multiple comparisons. Blue and green stars show Fisher's LSD post hoc test comparing sucralose (blue stars) or glucose (green stars) solution feeding to water drinking. All data shown are means with SEM. *$p < 0.05$. **$p < 0.01$. ***$p < 0.001$. ****$p < 0.0001$.

The online version of this article includes the following source data and figure supplement(s) for figure 1:

**Source data 1.** Statistical analysis for preferred temperatures (Tp).

**Figure supplement 1.** Compilation of refeeding experiments derived from *Figure 1D–G*.

starved flies touched glucose and sucralose food at similar frequencies and more frequently than water during the 30-min test period (*Figure 1H*, *Figure 1—source data 1*). The data suggest that flies are likely to feed on glucose and sucralose at similar rates. Therefore, we concluded that the differential effect of sucralose and glucose refeeding on temperature preference was not due to differences in feeding rate.

Furthermore, to confirm that sweet taste is more important than sugar structure, we instead used fructose, another simple sugar that contains sweetness and nutrients. Glucose and fructose are monosaccharides and a member of hexose and pentose, respectively. We tested fructose refeeding for 10 min and 1 hr. We found that 10 min of fructose refeeding resulted in full recovery of Tp from the starved state (*Figure 1G*, *Figure 1—figure supplement 1*, purple bar: statistics shown in red NS, *Figure 1—source data 1*), and 10 min and 1 hr fructose refeeding were significantly different from starved flies (*Figure 1G*, *Figure 1—figure supplement 1*: statistics shown in green stars, *Figure 1—source data 1*). The data suggest that sweet taste is more important than the structures of the sugar compounds.

## Activation of sweet taste neurons leads to warm preference

To determine how taste elicits a warm preference, we focused on the sweet gustatory receptors (Grs), which detect sweet taste. We used sweet Gr mutants and asked whether sweet Grs are involved in taste-evoked warm preference. Two different sweet Gr mutants, $Gr5a^{-/-}$; $Gr64a^{-/-}$ and $Gr5a^{-/-}$;;$Gr61a^{-/-}$, $Gr64a-f^{-/-}$, are known to reduce sugar sensitivity compared to the control (*Dahanukar et al., 2007*; *Dus et al., 2011*; *Yavuz et al., 2014*; *Fujii et al., 2015*). We found that sweet Gr mutant flies exhibited a normal starvation response in which the Tp of starved flies was lower than that of fed flies (*Figure 2A, B*, white and gray bars, statistics shown as green stars, *Figure 2—source data 1*). However, starved sweet Gr mutant flies did not increase Tp after 10 min sucralose refeeding (*Figure 2A, B*, blue bars, statistics shown as green and red stars, *Figure 2—source data 1*). These data suggest that sweet Grs are involved in taste-evoked warm preference.

Sweet Grs are expressed in the sweet Gr-expressing neurons (GRNs) located in the proboscis and forelegs (*Fujii et al., 2015*; *Thoma et al., 2016*). To determine whether sweet GRNs are involved in taste-evoked warm preference, we silenced all sweet GRNs. We expressed the inwardly rectifying K+ channel Kir2.1 (*uas-Kir*) (*Baines et al., 2001*) using *Gr64f-Gal4*, which is expressed in all sweet GRNs in the proboscis and forelegs (*Dahanukar et al., 2007*; *Fujii et al., 2015*; *Thoma et al., 2016*). Inactivation of all sweet GRNs showed a normal starvation response (*Figure 2C*, white and gray bars, statistics shown as green stars, *Figure 2—source data 1*). However, flies silencing all sweet GRNs failed to show a warm preference after 10 min of sucralose refeeding (*Figure 2C*, blue bar, statistics shown as green and red stars, *Figure 2—source data 1*). This phenotype was similar to the data obtained with the sweet Gr mutant strains (*Figure 2A, B*). On the other hand, control flies (*Gr64f-Gal4/+* and

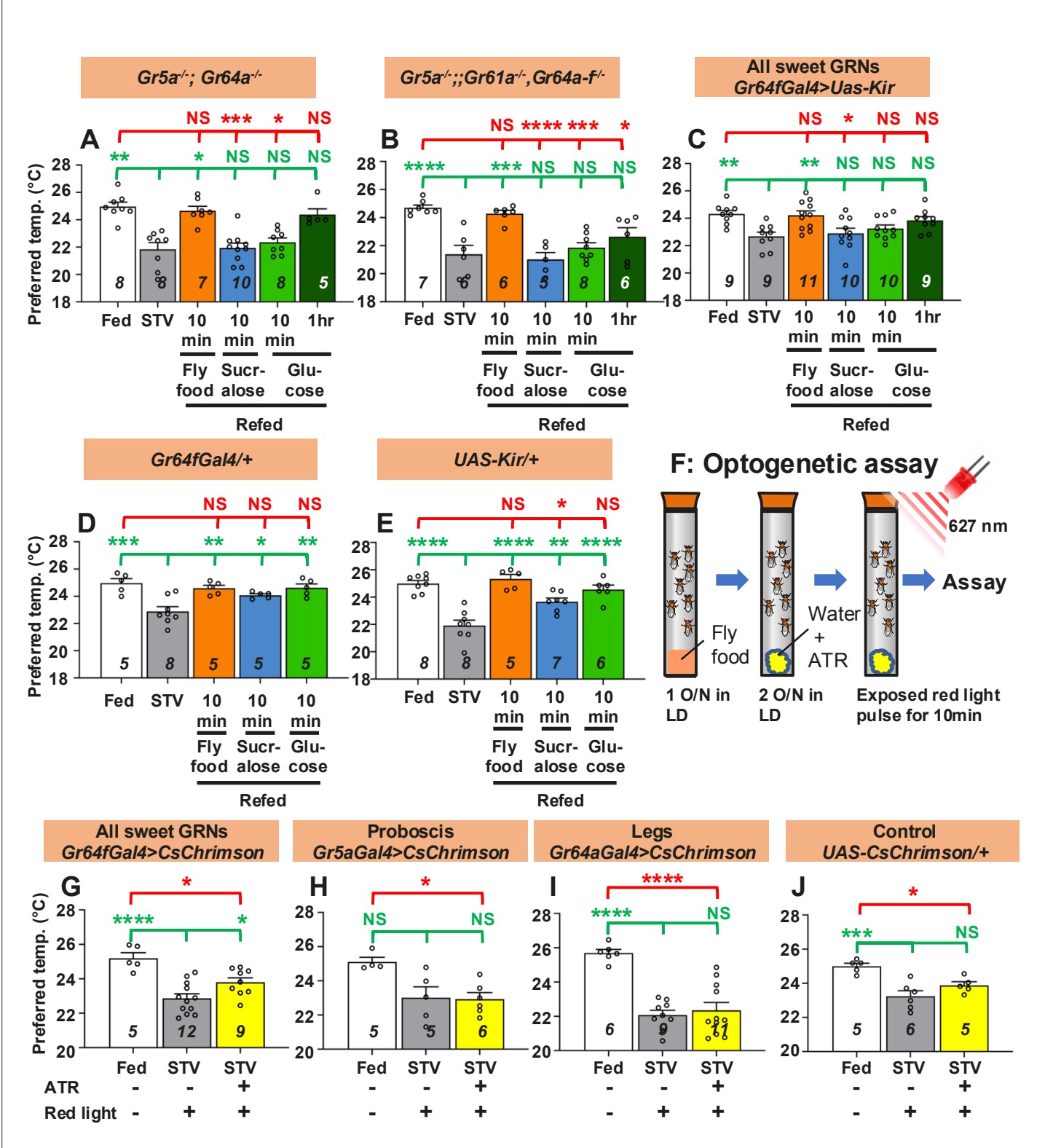

**Figure 2.** Gustatory neurons are essential for taste-evoked warm preference. (**A–E**) Comparison of preferred temperature (Tp) of flies between fed (white bar), starved (STV; gray bar), and refed (orange, blue, green, or dark green bar) states. Flies were starved for two overnights (ON) except for $Gr5a^{-/-};;Gr61a^{-/-}, Gr64a\text{-}f^{-/-}$ (1.5ON). Starved flies were refed with fly food for 10 min (fly food; orange bars), sucralose for 10 min (blue bars), or glucose for 10 min (green bars), or 1 hr (dark green bars). (**F**) Schematic of the optogenetic activation assay. (**G–J**) Comparison of Tp of flies between fed (white

*Figure 2 continued on next page*

*Figure 2 continued*

bar), starved (STV; gray bar), and starved with all-trans-retinal (ATR; yellow bars), which is the chromophore required for CsChrimson activation. Gustatory neurons in starved flies were excited by red light pulses (flashing on and off at 10 Hz) for 10 min. Starvation was performed for 2ON. Averaged fly distributions of flies in the temperature gradient for *Gr64fGal4>CsChrimson* are shown in **Figure 2—figure supplement 1B**. Behavioral experiments were performed on ZT4–7. Dots on each bar indicate individual Tp in assays. Numbers in italics indicate the number of trials. The Shapiro–Wilk test was used to test for normality. One-way ANOVA was used for statistical analysis. Red or green stars indicate Tukey's post hoc test compared between each experiment and the fed (red) or starved (green) condition. All data presented are means with SEM. *$p < 0.05$. **$p < 0.01$. ***$p < 0.001$. ****$p < 0.0001$. NS indicates not significant.

The online version of this article includes the following source data and figure supplement(s) for figure 2:

**Source data 1.** Statistical analysis for preferred temperatures (Tp).

**Figure supplement 1.** The distribution of temperature preference in each experimental condition.

*uas-Kir/+*) showed a normal starvation response and a taste-evoked warm preference (**Figure 2D, E**, gray and blue bars, statistics shown as green and red stars, **Figure 2—source data 1**). Thus, our data indicate that sweet GRNs are required for taste-evoked warm preference.

To further investigate whether activation of sweet GRNs induces a warm preference, we used the optogenetic approach, a red light sensitive channelrhodopsin, CsChrimson (**Klapoetke et al., 2014**; **Simpson and Looger, 2018**). Starved flies were given water containing 0.8 mM all-trans-retinal (ATR), the chromophore required for CsChrimson activation. These flies were not fed sucralose; instead, gustatory neurons in starved flies were excited by red light pulses (flashing on and off at 10 Hz) for 10 min (**Figure 2F**). In this case, although the flies were not refed, the gustatory neurons were artificially excited by CsChrimson activation so that we could evaluate the effect of excitation of sweet GRNs on taste-evoked warm preference.

CsChrimson was expressed in sweet GRNs in the proboscis and legs (all sweet GRNs) using *Gr64f-Gal4*. These flies showed a normal starvation response (**Figure 2G–J**, white and gray bars, statistics shown as green and red stars, **Figure 2—source data 1**). Excitation of all sweet GRNs by red light pulses elicited a warm preference, and Tp was intermediate between fed and starved flies, suggesting partial recovery (**Figure 2G**, yellow bar, statistics shown as green and red stars, **Figure 2—figure supplement 1**, and **Figure 2—source data 1**). However, neither excitation of Gr5a- (**Figure 2H**) nor Gr64a-expressing neurons (**Figure 2I**) induced a warm preference (yellow bars, **Figure 2—source data 1**). While *Gr64a-Gal4* is expressed only in the legs, *Gr5a-Gal4* is expressed in the proboscis and legs, but does not cover all sweet GRNs like *Gr64f-Gal4* (**Fujii et al., 2015**; **Thoma et al., 2016**). Notably, control flies (*UAS-CsChrimson/+*) did not show a warm preference to red light pulses with ATR application (**Figure 2J**). Taken together, our data suggest that excitation of all sweet GRNs results in a warm preference.

We next asked whether the sweet Grs contribute to the nutrient-induced warm preference. We found that all these starved flies did not increase Tp after 10 min of glucose intake (**Figure 2A–C**, green bars, statistics shown as green and red stars, **Figure 2—source data 1**). All control flies showed normal responses to 10 min of glucose refeeding (**Figure 2D, E**, green bars, **Figure 2—source data 1**). The data suggest that the sweet Grs which we tested are potentially expressed in tissues/neurons required for internal nutrient sensing. Notably, we found that flies increased Tp after 10 min of refeeding with fly food containing carbohydrate, fat, and protein (**Figure 2A–C**, orange bars, statistics shown as green and red stars, **Figure 2—source data 1**). All control flies showed normal responses to 10 min of fly food intake (**Figure 2D, E**, orange bars, **Figure 2—source data 1**). The data suggest that gustatory neurons are required for warm preference in carbohydrate refeeding, but not for other nutrients such as fat or protein (see Discussion). Because flies have sensory neurons that detect fatty acids (**Masek and Keene, 2013**; **Ahn et al., 2017**; **Brown et al., 2021**) or amino acids (**Croset et al., 2016**; **Ganguly et al., 2017**; **Chen and Dahanukar, 2017**; **Steck et al., 2018**), these neurons may drive the response to fly food intake. This is likely why the sweet-insensitive flies can still recover after eating fly food (**Figure 2A–C**, orange bars).

## The temperature-sensing neurons are involved in taste-evoked warm preference

The warm-sensing neurons, anterior cells (ACs), and the cold-sensing *R11F02-Gal4*-expressing neurons control temperature preference behavior (**Umezaki et al., 2018**; **Hamada et al., 2008**; **Ni**

*et al., 2016*). Small ectotherms such as *Drosophila* set their Tp to avoid noxious temperatures using temperature information from cold- and warm-sensing neurons (*Sayeed and Benzer, 1996*; *Dillon et al., 2009*; *Hamada et al., 2008*). We have previously shown that starved flies choose a lower Tp, the so-called hunger-driven lower Tp (*Umezaki et al., 2018*). ACs control the hunger-driven lower Tp, but cold-sensing *R11F02-Gal4*-expressing neurons do not (*Umezaki et al., 2018*). ACs express transient receptor potential A1 (TrpA1), which responds to a warm temperature >25°C (*Hamada et al., 2008*; *Tang et al., 2013*). The set point of ACs in fed flies, which is ~25°C, is lowered in starved flies. Therefore, the lower set point of ACs corresponds to the lower Tp in starved flies.

First, we asked whether ACs are involved in taste-evoked warm preference. Because the ACs are important for the hunger-driven lower Tp (*Umezaki et al., 2018*), the AC-silenced flies did not show a significant difference in Tp between fed and starved conditions for only one overnight of starvation (*Umezaki et al., 2018*). Therefore, we first extended the starvation time to two overnights so that the AC-silenced flies showed a significant difference in Tp between fed and starved conditions (*Figure 3A*, white and gray bars, statistics shown as green and red stars, *Figure 3—source data 1*). Importantly, longer periods of starvation do not affect the ability of $w^{1118}$ flies to recover (*Figure 3— figure supplement 1*, *Figure 3—figure supplement 1—source data 1*).

To examine whether ACs regulate taste-evoked warm preference, we refed sucralose to AC-silenced flies for 10 min. We found that the Tp of the refed flies was still similar to that of the starved flies (*Figure 3A*, blue bar, statistics shown as green and red stars, *Figure 3—source data 1*), indicating that sucralose refeeding could not restore Tp and that ACs are involved in taste-evoked warm preference. Significantly, even when the AC-silenced flies were starved for two overnights, they were able to recover Tp to the normal food for 10 min and to glucose for 1 hr (*Figure 3A*, orange and green bars, statistics shown as green and red stars, *Figure 3—source data 1*), suggesting that the starved AC-silenced flies were still capable of recovery.

Other temperature-sensing neurons involved in temperature preference behavior are cold-sensing *R11F02-Gal4*-expressing neurons (*Umezaki et al., 2018*; *Ni et al., 2016*). To determine whether *R11F02-Gal4*-expressing neurons are involved in taste-evoked warm preference, we silenced *R11F02-Gal4*-expressing neurons using *uas-Kir*. We found that the flies showed a significant difference in Tp between fed and starved conditions, but the flies did not show a warm preference upon sucralose refeeding (*Figure 3B*, gray and blue bars, statistics shown as green and red stars, *Figure 3—source data 1*). As controls, $TrpA1^{SH}$-Gal4/+, R11F02-Gal4/+, and uas-Kir/+ flies showed normal starvation response and taste-evoked warm preference (*Figure 3C–E*).

To further ensure the results, we used optogenetics to artificially excite warm and cold neurons with $TrpA1^{SH}$-Gal4 and R11F02-Gal4, respectively, by red light pulses for 10 min. We compared the Tp of starved flies with and without ATR under red light. Tp of starved flies with ATR was significantly increased (*Figure 3F, G*, yellow bars, statistics shown as black stars, *Figure 3—source data 1*) compared to those without ATR (*Figure 3F, G*, gray bars, *Figure 3—source data 1*). Therefore, these data indicate that ACs and *R11F02-Gal4*-expressing neurons are required for taste-evoked warm preference.

Next, we asked whether temperature-sensitive neurons contribute to the nutrient-induced warm preference. We used the warm- or cold-neuron-silenced flies ($TrpA1^{SH}$-Gal4 or R11F02-Gal4>uas-Kir) and found that all these starved flies did not increase Tp after 10 min glucose intake (*Figure 3A, B*, green bars, *Figure 3—source data 1*), but increased Tp after 10 min refeeding with fly food containing carbohydrate, fat, and protein (*Figure 3A, B*, orange bars, *Figure 3—source data 1*). Notably, AC-silenced flies increased Tp after 1 hr glucose intake (*Figure 3A*, green bars, *Figure 3—source data 1*). All control flies showed normal responses to both 10 min glucose refeeding and fly food intake (*Figure 3C–E*, green bars, *Figure 3—source data 1*). The data suggest that temperature-sensing neurons are required for warm preference in carbohydrate refeeding, but not in other foods such as fat or protein (see Discussion).

## Olfaction is possibly involved in a warm preference for hungry flies

We also investigated the potential effects of olfaction. We used mutants of the odorant receptor co-receptor, Orco ($Orco^1$), which has an olfactory defect (*Larsson et al., 2004*). We found that the flies showed a significant difference in Tp between fed and starved conditions, but the flies did not show a warm preference upon sucralose refeeding (*Figure 3H*, gray and blue bars, statistics shown as green

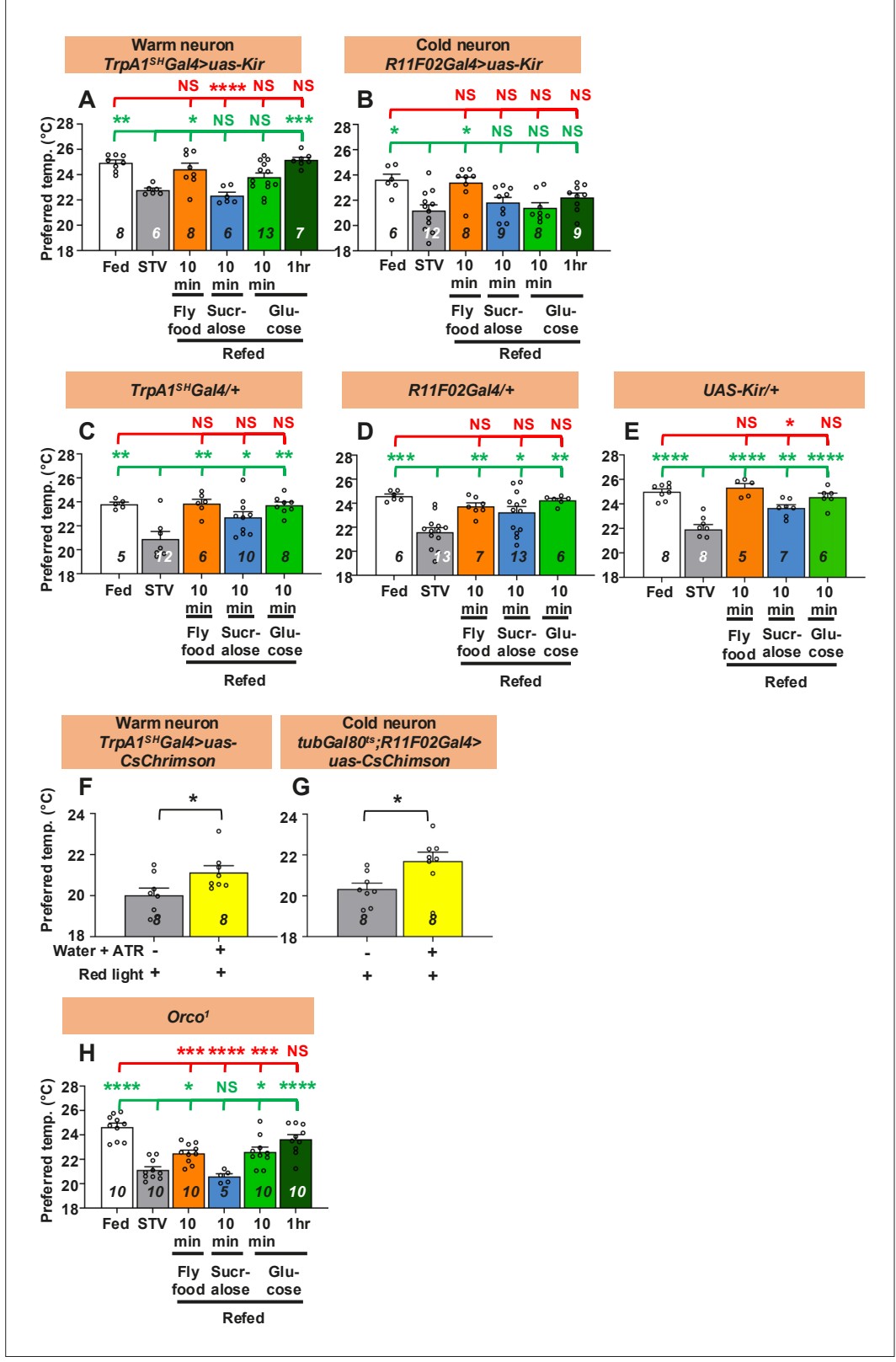

**Figure 3.** Both warm and cold temperature-sensing neurons are involved in taste-evoked warm preference. (**A–E, H**) Comparison of preferred temperature (Tp) of flies between fed (white bar), starved (STV; gray bar), and refed (orange, blue, green, or dark green bar) conditions. Starvation was applied for two overnights (ON). Starved flies were refed with fly food for 10 min (orange bar), sucralose for 10 min (blue bar), or glucose for 10 min (green bar) or

*Figure 3 continued on next page*

*Figure 3 continued*

1 hr (dark green bar). The Shapiro–Wilk test was used to test for normality. One-way ANOVA or Kruskal–Wallis test was used for statistical analysis. Red or green stars indicate Tukey's post hoc test or Dunn's test compared between each experiment and the fed (red) or starved (green) condition, respectively. (**F, G**) Comparison of Tp between starved (STV; gray bar) and all-trans-retinal (ATR; yellow bar) starved flies. Starvation was performed for 2ON. Warm neurons (**F**) or cold neurons (**G**) in starved flies expressed CsChrimson, which was excited by red light pulses for 10 min. The Shapiro–Wilk test was performed to test for normality. Student's *t*-test or Kolmogorov–Smirnov test was used for statistical analysis. (**G**) *tubGal80$^{ts}$; R11F02-Gal4>uas-CsChrimson* flies were reared at 18°C, and emerged adults were collected and stored at 29°C. See Materials and methods for details. These behavioral experiments were performed on ZT4–7. The dots on each bar indicate individual Tp in the assays. Numbers in italics indicate the number of experiments. All data shown are means with SEM. *$p < 0.05$. **$p < 0.01$. ***$p < 0.001$. ****$p < 0.0001$. NS indicates not significant.

The online version of this article includes the following source data and figure supplement(s) for figure 3:

**Source data 1.** Statistical analysis for preferred temperatures (Tp).

**Figure supplement 1.** Duration of starvation is unlikely to affect the ability to recover.

**Figure supplement 1—source data 1.** Statistical analysis for preferred temperatures (Tp).

and red stars, *Figure 3—source data 1*). We found that all of these starved flies increased Tp after 10 min of glucose or fly food intake and showed a full recovery after 1 hr of glucose intake (*Figure 3H*, orange and green bars, statistics shown as green and red stars, *Figure 3—source data 1*). The data suggest that olfaction may be involved in the warm preference in sucralose refeeding.

## Internal state influences taste-evoked warm preference in hungry flies

Internal state strongly influences motivation to feed. However, how internal state influences starving animals to exhibit a food response remains unclear. Hunger represents the food-deficient state in the body, which induces the release of hunger signals such as NPY. NPY promotes foraging and feeding behavior in mammals and flies (*Nässel and Wegener, 2011*). While intracerebroventricular injection of NPY induces the CPR (*Geoghegan et al., 1993*), injection of NPY antagonists suppresses CPR in dogs, suggesting that NPY is a regulator of CPR in mammals (*Lee et al., 1994*). Therefore, we first focused on neuropeptide F (NPF) and small neuropeptide F (sNPF), which are the *Drosophila* homolog and ortholog of mammalian NPY, respectively (*Nässel and Wegener, 2011*), and asked whether they are involved in taste-evoked warm preference. In *NPF* mutant (*NPF$^{-/-}$*) or *sNPF* hypomorph (*sNPF hypo*) mutant, we found that the Tp of fed and starved flies were significantly different, showing a normal starved response (*Figure 4A, B*: white and gray bars, statistics shown as red stars, *Figure 4—source data 1*). However, they failed to show a taste-evoked warm preference after 10 min of sucralose refeeding (*Figure 4A, B*: blue bars, statistics shown as green and red stars, *Figure 4—source data 1*). Thus, NPF and sNPF are required for taste-evoked warm preference after sucralose refeeding.

We next asked whether NPF and sNPF are involved in the nutrient-induced warm preference during glucose refeeding. We found that starved *NPF$^{-/-}$* mutants significantly increased Tp after 1 hr of glucose refeeding (*Figure 4A*, dark green bar, statistics shown as green and red stars, *Figure 4—source data 1*). We also found that starved *sNPF hypo* mutants significantly increased Tp after 10 min and 1 hr of glucose refeeding (*Figure 4B*, green and dark green bars, statistics shown as green and red stars, *Figure 4—source data 1*). In addition, both *NPF$^{-/-}$* and *sNPF hypo* mutants increased Tp after 10 min of fly food refeeding (*Figure 4A, B*, orange bars, statistics shown as green and red stars, *Figure 4—source data 1*). However, when these flies were fed both glucose and fly food, they did not reach the same Tp as the fed flies. These data suggest that NPF and sNPF also play a role in the modulation of nutrient-induced warm preference.

## Factors involving hunger regulate taste-evoked warm preference

Based on the above results, we hypothesized that hunger signals might be involved in taste-evoked warm preference. To test this hypothesis, we focused on several factors involved in the hunger state. The major hunger signals (*Lin et al., 2019*), diuretic hormone 44 (DH44), and adipokinetic hormone (AKH) are the mammalian corticotropin-releasing hormone homolog (*Dus et al., 2015*) and the functional glucagon homolog, respectively (*Lee and Park, 2004*; *Kim and Rulifson, 2004*). We found that DH44- or AKH-expressing neuron silenced flies failed to show a taste-evoked warm preference after

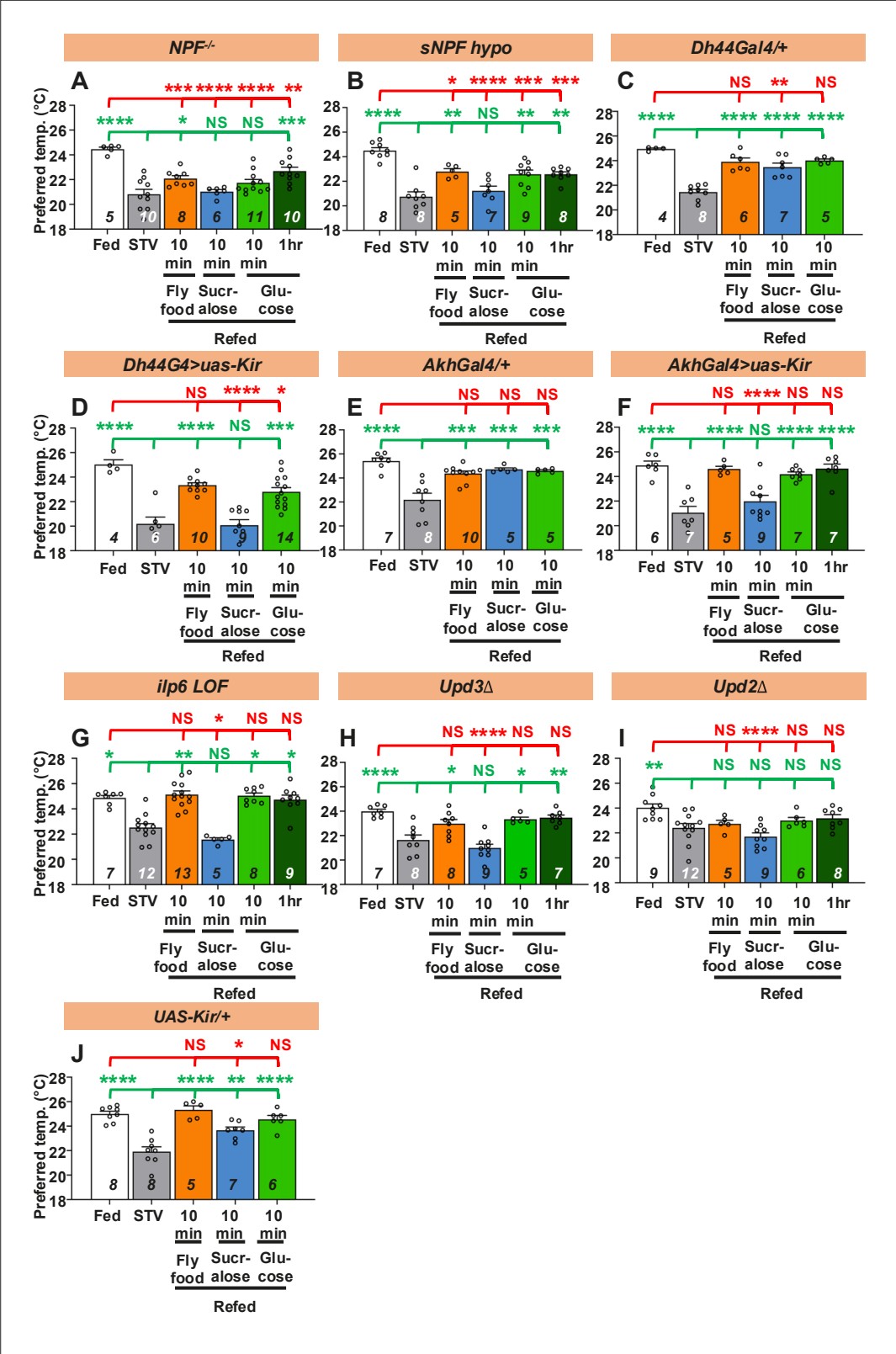

**Figure 4.** Hunger signals are involved in taste-evoked warm preference. (**A-J**) Comparison of preferred temperature (Tp) of flies between fed (white bar), starved (STV; gray bar), and refed states (orange, blue, green, or dark green). Flies were starved for two overnights (ON) except for *ilp6* mutant flies (3ON). Starved flies were refed with fly food (orange bar), sucralose (blue bar), or glucose (green bar) for 10 min. These behavioral experiments

*Figure 4 continued on next page*

*Figure 4 continued*

were conducted on ZT4–7. Dots on each bar indicate individual Tp in the assays. Numbers in italics indicate the number of trials. Shapiro–Wilk test was performed for normality test. One-way ANOVA was used for statistical analysis. Red or green stars indicate Tukey's post hoc test comparing between each experiment to the fed (red) or starved (green) condition, respectively. All data presented are means with SEM. $*p < 0.05$. $**p < 0.01$. $***p < 0.001$. $****p < 0.0001$. NS indicates not significant.

The online version of this article includes the following source data and figure supplement(s) for figure 4:

**Source data 1.** Statistical analysis for preferred temperatures (Tp).

**Figure supplement 1.** Number of touches to water, sucralose, and glucose using *yw*, *per[01]*, and *tim[01]* flies.

**Figure supplement 1—source data 1.** Statistical analysis for preferred temperatures (Tp).

sucralose refeeding. However, they were able to increase Tp after 10 min glucose or fly food refeeding (*Figure 4D, F*, green and orange bars, statistics shown as green and red stars, *Figure 4—source data 1*), suggesting a normal nutrient-induced warm preference. The control flies (*Dh44-Gal4/+*, *Akh-Gal4/+*, and *UAS-Kir/+*) showed a normal warm preference to sucralose (*Figure 4C, E, J*, blue bars, statistics shown as green and red stars, *Figure 4—source data 1*), glucose (*Figure 4C, E, J*, green bars, statistics shown as green and red stars, *Figure 4—source data 1*), and normal fly food refeeding (*Figure 4C, E, J*, orange bars, statistics shown as green and red stars, *Figure 4—source data 1*). The data suggest that DH44 or AKH neurons are required for taste-evoked warm preference, but not for nutrient-induced warm preference.

Insulin-like peptide 6 (Ilp6) is a homolog of mammalian insulin-like growth factor 1 (IGF1), and *ilp6* mRNA expression is increased in starved flies (*Okamoto et al., 2009*; *Slaidina et al., 2009*; *Bai et al., 2012*). Because Ilp6 is important for the hunger-driven lower Tp (*Umezaki et al., 2018*), the *ilp6* mutants did not show a significant difference in Tp between fed and starved conditions for only one overnight starvation. Therefore, we first extended the starvation time to three overnights. We found that the *ilp6* loss-of-function (*ilp6 LOF*) mutant failed to show a taste-evoked warm preference after sucralose refeeding (*Figure 4G*, blue bars, statistics shown as green and red stars, *Figure 4—source data 1*), but did show a nutrient-induced warm preference after glucose refeeding (*Figure 4G*, green bars, statistics shown as green and red stars, *Figure 4—source data 1*). These data suggest that Ilp6 is required for taste-evoked warm preference but not for nutrient-induced warm preference after glucose or fly food refeeding.

Unpaired3 (Upd3) is a *Drosophila* cytokine that is upregulated under nutritional stress (*Woodcock et al., 2015*). We found that the *upd3* mutants failed to show a taste-evoked warm preference after sucralose refeeding (*Figure 4H*, blue bar, statistics shown as green and red stars, *Figure 4—source data 1*), but showed a nutrient-induced warm preference after glucose or normal food refeeding (*Figure 4H*, green and orange bars, statistics shown as green and red stars, *Figure 4—source data 1*). These data suggest that Upd3 is required for taste-evoked warm preference, but not for nutrient-induced warm preference after glucose or normal food refeeding.

We also examined the role of the satiety factor Unpaired2 (Upd2), a functional leptin homolog in flies (*Rajan and Perrimon, 2012*), in taste-evoked warm preference. The *upd2* mutants failed to show a warm preference after refeeding of sucralose, glucose or fly food (*Figure 4I*, blue, green and orange bars, statistics shown as green and red stars, *Figure 4—source data 1*). Thus, our data suggest that factors involved in the hunger state are required for taste-evoked warm preference.

## Flies show a taste-evoked warm preference at all times of the day

Animals anticipate feeding schedules at a time of day that is tightly controlled by the circadian clock (*Chaix et al., 2019*). Flies show a rhythmic feeding pattern: one peak in the morning (*Ro et al., 2014*) or two peaks in the morning and evening (*Xu et al., 2008*). Because food cues induce a warm preference, we wondered whether feeding rhythm and taste-evoked warm preference are coordinated. If so, they should show a parallel phenotype.

Since flies exhibit one of the circadian outputs, the TPR (*Kaneko et al., 2012*), Tp gradually increases during the day and peaks in the evening. First, we tested starvation responses at Zeitgeber time (ZT)1–3, 4–6, 7–9, and 10–12 under light and dark (LD) conditions, with flies being offered only water for 24 hr prior to the experiments at each time point. We found that both *w[1118]* and *yellow[1]*

*white[1]* (*y[1]w[1]*) flies had higher Tp in the fed state and lower Tp in the starved state at all times of the day (*Figure 5A, B*: black and gray lines; *Figure 5—source data 1*). Next, we refed sucralose to starved flies at all time points tested and examined taste-evoked warm preference. While starved *y[1]w[1]* flies showed a taste-evoked warm preference at all time points (*Figure 5B1*, gray and blue lines, *Figure 5—source data 1*), starved *w[1118]* flies showed a significant taste-evoked warm preference at ZT4–6 and 10–12 (*Figure 5A1*, gray and blue lines, *Figure 5—source data 1*).

Because starved *w[1118]* flies showed an advanced phase shift of TPR with a peak at ZT7–9 (*Figure 5A*, gray line), it is likely that the highest Tp simply masks the taste-evoked warm preference at ZT7–9. We also focused on nutrient- (carbohydrate-) induced warm preference. Starved *w[1118]* and *y[1]w[1]* flies successfully increased Tp after 10 min of glucose refeeding (*Figure 5A2, B2*, green line, *Figure 5—source data 1*). Glucose refeeding for 1 hr resulted in Tp similar to that of fed *w[1118]* flies (*Figure 5A3*, dark green line). Because the feeding rhythm peaks in the morning or morning/evening (*Ro et al., 2014*; *Xu et al., 2008*), our data suggest that the feeding rhythm and taste-evoked warm preference do not occur in parallel.

## Circadian clock genes are required for taste-evoked warm preference, but not for nutrient-induced warm preference

We asked whether the circadian clock is involved in taste-evoked warm preference. We used clock gene null mutants, *period[01]* (*per[01]*) and *timeless[01]* (*tim[01]*). Although they showed significant starvation responses (*Figure 5C, D*, black and gray lines, *Figure 5—source data 1*), neither starved *per[01]* nor *tim[01]* mutants could show taste-evoked warm preference upon sucralose refeeding (*Figure 5C1, D1*, blue lines, *Figure 5—source data 1*). Nevertheless, they fully recovered upon glucose refeeding in LD at any time of day (*Figure 5C2, D2*, green lines, *Figure 5—source data 1*). Therefore, our data suggest that clock genes are required for taste-evoked warm preference, but not for nutrient-induced warm preference.

However, starved *per[01]* and *tim[01]* mutants may eat sucralose less frequently than glucose, which could result in a failure to show a taste-evoked warm preference. Therefore, we examined how often starved *per[01]* and *tim[01]* mutants touched glucose, sucralose, or water during the 30 min using FLIC assays (*Ro et al., 2014*; *Figure 4—figure supplement 1*). Interestingly, starved *per[01]* and *tim[01]* mutants touched water significantly more often than glucose or sucralose (*Figure 4—figure supplement 1*, *Figure 4—figure supplement 1—source data 1*). Although starved *tim[01]* flies touched glucose slightly more than sucralose for only 10 min, this phenotype is not consistent with *per[01]* and *w[1118]* flies. However, these mutants still showed a similar Tp pattern for sucralose and glucose refeeding (*Figure 5C, D*). The results suggest that although the *tim[01]* flies can eat sufficient amount of sucralose over glucose, their food intake does not affect the Tp behavioral phenotype. Thus, we conclude that in *per[01]* and *w[1118]* flies, the differential response between taste-evoked and nutrient-induced warm preferences is not due to feeding rate.

## Discussion

When animals are hungry, sensory detection of food (sight, smell, or chewing) initiates digestion even before the food enters the stomach. The food-evoked responses are also observed in thermogenesis, heart rate, and respiratory rate in mammals (*LeBlanc, 2000*; *LeBlanc and Cabanac, 1989*; *Nederkoorn et al., 2000*). These responses are referred to as the CPR and contribute to the physiological regulation of digestion, nutrient homeostasis, and daily energy homeostasis (*Zafra et al., 2006*). While starved flies show a cold preference, we show here that the food cue, such as the excitation of gustatory neurons, triggers a warm preference, and the nutritional value triggers an even higher warm preference. Thus, when flies exit the starvation state, they use a two-step approach to recovery, taste-evoked and nutrient-induced warm preferences. The taste-evoked warm preference in *Drosophila* may be a physiological response potentially equivalent to CPR in mammals. Furthermore, we found that internal needs, controlled by hunger signals and circadian clock genes, influence taste-evoked warm preference. Thus, we propose that the taste-evoked response plays an important role in recovery and represents another layer of regulation of energy homeostasis.

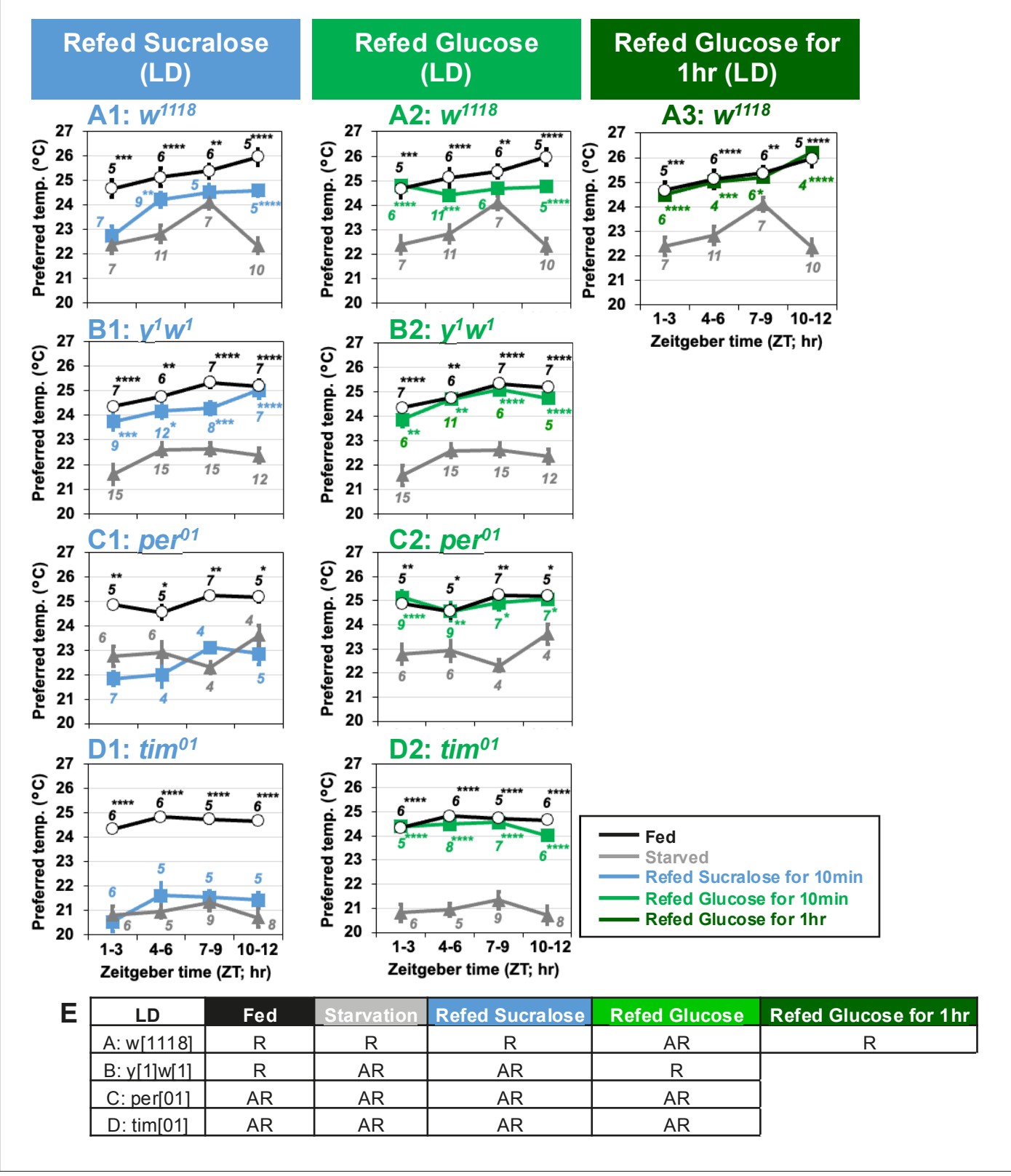

**Figure 5.** Clock genes are involved in taste-evoked warm preference. (**A–D**) Comparison of preferred temperature (Tp) of flies between fed (white circles), starved (STV; gray triangles), and refed (blue, green, or dark green squares) states. Flies were starved for 24 hr. Starved flies were refed with sucralose blue squares (**A1–D1**) or glucose for 10 min green squares (**A2–D2**) or 1 hr dark green squares (**A3**). After the 10 min or 1 hr refeeding, the temperature preference behavior assays were performed immediately at ZT1–3, ZT4–6, ZT7–9, and ZT10–12 in LD. The Shapiro–Wilk test was performed

*Figure 5 continued on next page*

*Figure 5 continued*

to test for normality. One-way ANOVA or Kruskal–Wallis test was used for statistical analysis. Stars indicate Tukey's post hoc test or Dunn's test compared between each experiment and the starved condition at the same time point. All data presented are means with SEM. *p < 0.05. **p < 0.01. ***p < 0.001. ****p < 0.0001. (**E**) Comparison of Tp during the daytime in each feeding state. One-way ANOVA or Kruskal–Wallis test was used for statistical analysis between ZT1–3 and ZT7–9 or ZT10–12, respectively. R and AR indicate rhythmic and arrhythmic, respectively, during the daytime.

The online version of this article includes the following source data for figure 5:

**Source data 1.** Statistical analysis for preferred temperatures (Tp).

## Tp is determined by the taste cue

Starved flies increase Tp in response to a nutrient-free taste cue (*Figure 1E*, *Figure 1—figure supplement 1*), resulting in a taste-evoked warm preference. We showed that silencing of ACs or cold neurons caused a loss of taste-evoked warm preference (*Figure 3A–E*), and that excitation of ACs or cold neurons induced a taste-evoked warm preference (*Figure 3F, G*). The data suggest that both warm and cold neurons are important for taste-evoked warm preference: while ACs are required for the hunger-driven lower Tp (*Umezaki et al., 2018*), both ACs and cold neurons are likely to be important for this taste-evoked warm preference.

The hunger-driven lower Tp is a slower response because starvation gradually lowers their Tp (*Umezaki et al., 2018*). In contrast, the taste-evoked warm preference is a rapid response. Once ACs and cold neurons are directly or indirectly activated, starved flies quickly move to a warmer area. This is interesting because even when ACs and cold neurons are activated by warm and cold, respectively, the activation of these neurons causes a warm preference. Given that the sensory detection of food

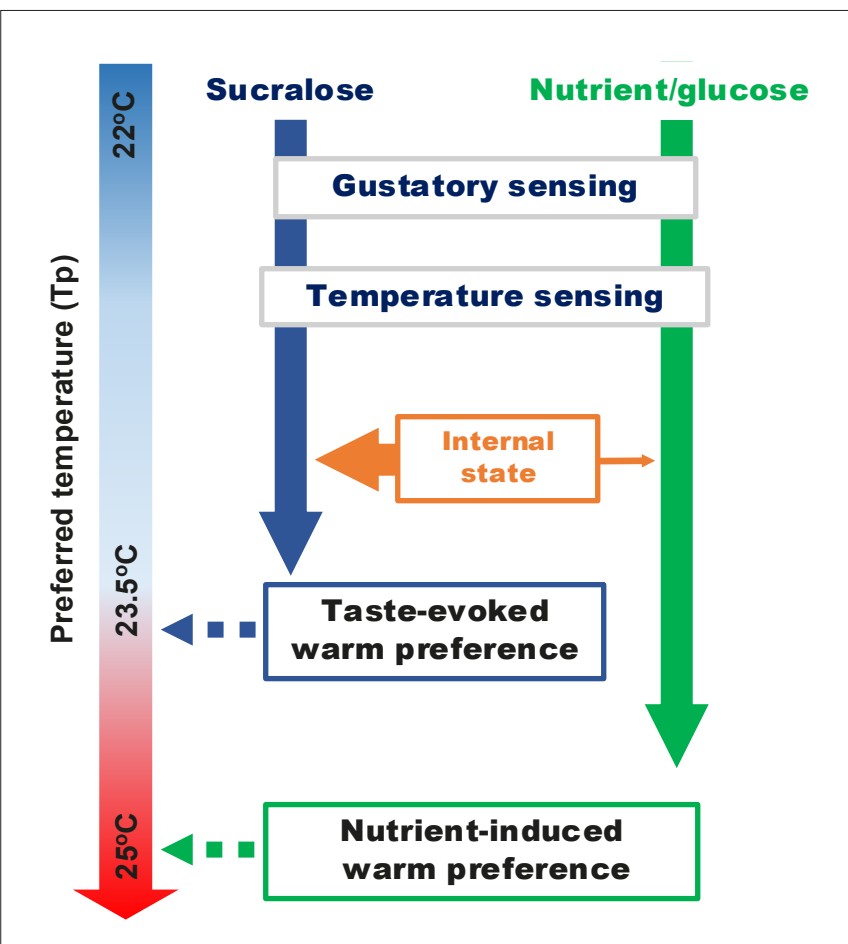

**Figure 6.** A schematic diagram of the recovery of preferred temperature (Tp) by taste-evoked warm preference (blue arrows) and nutrient-induced warm preference (green arrows).

(sight, smell, or chewing food) triggers CPR, the activation of these sensory neurons may induce CPR. Tp in sucralose-refed hungry flies is between that of fed and starved flies (*Figures 2 and 3*), making it difficult to detect the smaller temperature differences using the calcium imaging experiments. Therefore, we speculate that both ACs and cold neurons may facilitate rapid recovery from starvation so that flies can quickly return to their preferred temperature – body temperature – to a normal state.

## Internal state influences taste-evoked warm preference

We show that mutants of genes involved in hunger and the circadian clock fail to show taste-evoked warm preference, suggesting that hunger and clock genes are important for taste-evoked warm preference. At a certain time of day, animals are hungry for food (*Ro et al., 2014*; *Xu et al., 2008*). Thus, the hungry state acts as a gatekeeper, opening the gate of the circuits when hungry flies detect the food information that leads to taste-evoked warm preference (*Figures 4 and 6*, blue arrow). While most of the hunger signals we focused on are important for taste-evoked warm preference, some hunger signals are also required for both taste-evoked and nutrient-induced warm preferences (*Figure 4*). Notably, sensory signals contribute to both taste-evoked and nutrient-induced warm preferences (*Figures 2, 3, and 6*, blue and green arrows). Thus, taste-evoked warm preference and nutrient-induced warm preference differ at the internal state level, but not at the sensory level. This idea is analogous to appetitive memory formation. Sweet taste and nutrients regulate the different layers of the memory formation process. Recent evidence suggests that the rewarding process can be subdivided; the sweet taste is for short-term memory and the nutrient is for long-term memory (*Musso et al., 2017*; *Gruber et al., 2013*; *Huetteroth et al., 2015*). The data suggest that the taste-evoked response functions differently from the nutrient-induced response. Therefore, taste sensation is not just the precursor to nutrient sensing/absorption, but plays an essential role in the rapid initiation of a taste-evoked behavior that would help the animal survive.

## How do hunger signals or clock genes contribute to taste-evoked warm preference?

The hunger signaling hormones/peptides studied in this project are important for taste modulation. For example, mammalian NPY and its *Drosophila* homolog NPF modulate the output of taste signals (*Wang et al., 2016*; *Inagaki et al., 2014*; *Herness and Zhao, 2009*). The AKH receptor is expressed in a subset of gustatory neurons that may modulate taste information for carbohydrate metabolism (*Bharucha et al., 2008*). Therefore, the hunger signals are likely to modulate the downstream of the sensory neurons, which may result in a taste-evoked warm preference.

Circadian clock genes control and coordinate the expression of many clock-controlled genes in the body (*Giebultowicz, 2018*; *Mazzoccoli et al., 2012*; *Lin et al., 2002*; *Bozek et al., 2009*). Therefore, we expect that the absence of clock genes will disrupt the molecular and neural networks of homeostasis, including metabolism, that are essential for animal life. For example, taste neurons express clock genes, and impaired clock function in taste neurons disrupts daily rhythms in feeding behavior (*Chatterjee et al., 2010*). Temperature-sensing neurons transmit hot or cold temperature information to central clock neurons (*Tang et al., 2017*; *Alpert et al., 2020*; *Marin et al., 2020*; *Yadlapalli et al., 2018*; *Jin et al., 2021*). Therefore, the disrupted central clock in clock mutants may respond imprecisely to temperature signals. There are many possible reasons why the lack of clock gene expression in the brain is likely to cause abnormal taste-evoked warm preference.

In addition, hunger signals may contribute to the regulation of circadian output. DH44 is located in the dorsomedial region of the fly brain, the pars intercerebralis, and DH44-expressing neurons play a role in the output pathway of the central clock (*King et al., 2017*; *Barber et al., 2021*). Insulin-producing cells (IPCs) are also located in addition to DH44-expressing neurons (*Cao and Brown, 2001*; *Brogiolo et al., 2001*; *Ikeya et al., 2002*; *Rulifson et al., 2002*). IPCs receive a variety of information, including circadian (*Barber et al., 2021*; *Cavanaugh et al., 2014*) and metabolic signals (*Okamoto et al., 2009*; *Slaidina et al., 2009*; *Bai et al., 2012*; *Rajan and Perrimon, 2012*). and then transduce the signals downstream to release Ilps. Both Upd2 and Ilp6, which are expressed in the fat body respond to metabolic states and remotely regulate Ilp expression (*Okamoto et al., 2009*; *Slaidina et al., 2009*; *Bai et al., 2012*; *Rajan and Perrimon, 2012*). Insect fat body is analogous to the fat tissues and liver in the vertebrates (*Li et al., 2019*; *Arrese and Soulages, 2010*). Therefore,

each hunger signal may have its specific function for taste-evoked warm preference. Further studies are needed to describe the entire process.

## Taste-evoked warm preference may be CPR in flies

The introduction of food into the body disrupts the internal milieu, so CPR is a necessary process that helps animals prepare for digestion. Specifically, in mammals, taste leads to an immediate increase in body temperature and metabolic rate. Starvation results in lower body temperatures, and chewing food, even before it enters the stomach, triggers a rapid increase in heat production, demonstrating CPR in thermogenesis (*LeBlanc, 2000*; *LeBlanc and Cabanac, 1989*).

Starved flies have a lower Tp (*Umezaki et al., 2018*). Because *Drosophila* is a small ectotherm, the lower Tp indicates a lower body temperature (*Stevenson, 1985a*; *Stevenson, 1985b*). Even when the flies do not receive food, the sweet taste and the excitation of sweet neurons induce starved flies to show a warm preference, which eventually leads to a warmer body temperature. In fact, CPR is known to be influenced by smell as well in mammals (*Chen and Knight, 2016*). We have shown in flies that olfactory mutants fail to show a warm preference when refed sucralose (*Figure 3H*). Starvation leads to lower body temperatures, and food cues, including taste and odor, rapidly induce a rise in body temperature before food enters the body. Thus, the taste-evoked warm preference in *Drosophila* may be a physiological response equivalent to one of the CPRs observed in mammals.

## CPR is essential because both starved mammals and starved flies must rapidly regulate their body temperature to survive

As soon as starved flies taste food, the sensory signals trigger CPR. They can move to a warmer place to prepare to raise their body temperature (*Figure 6*, blue arrows). CPR may allow flies to choose a more hospitable place to restore their physiological state and allow for a higher metabolism, and eventually move on to the next step, such as foraging and actively seeking a mate before competitors arrive. Thus, CPR may be a strategy for the fly's survival. Similarly, starvation or malnutrition in mammals leads to lower body temperatures (*Piccione et al., 2002*; *Sakurada et al., 2000*; *Cintron-Colon et al., 2017*), and biting food triggers heat production, which is CPR (*LeBlanc, 2000*; *LeBlanc and Cabanac, 1989*; *LeBlanc et al., 1984*). Thus, while starvation in both flies and mammals leads to lower body temperatures, food cues initiate CPR by increasing body temperature and nutrient intake, resulting in full recovery from starvation. Our data suggest that *Drosophila* CPR may be a physiological response equivalent to CPR observed in other animals. Thus, *Drosophila* may shed new light on the regulation of CPR and provide a deeper understanding of the relationship between CPR and metabolism.

# Materials and methods

**Key resources table**

| Reagent type (species) or resource | Designation | Source or reference | Identifiers | Additional information |
|---|---|---|---|---|
| Genetic reagent (*D. melanogaster*) | $w^{1118}$ | Bloomington *Drosophila* Stock Center | BDSC:5905; RRID:BDSC_5905 | |
| Genetic reagent (*D. melanogaster*) | $y^1 w^1$ | Bloomington *Drosophila* Stock Center | BDSC:1495; RRID:BDSC_1495 | |
| Genetic reagent (*D. melanogaster*) | EP5Δ; Gr64a$^1$ | Dr. Anupama Dahanukar | PMID:17988633 | |
| Genetic reagent (*D. melanogaster*) | R1; Gr5a-LexA; +; ΔGr61a, ΔGr64a-f | Dr. Hubert Amrein | PMID:25984594; PMID:25702577 | |
| Genetic reagent (*D. melanogaster*) | Gr64f-Gal4 | Bloomington *Drosophila* Stock Center | BDSC:57669; RRID:BDSC_57669 | |
| Genetic reagent (*D. melanogaster*) | Gr5a-Gal4 | Bloomington *Drosophila* Stock Center | BDSC:57992; RRID:BDSC_57992 | |
| Genetic reagent (*D. melanogaster*) | Gr64a-Gal4 | Bloomington *Drosophila* Stock Center | BDSC:57661; RRID:BDSC_57661 | |

*Continued on next page*

*Continued*

| Reagent type (species) or resource | Designation | Source or reference | Identifiers | Additional information |
|---|---|---|---|---|
| Genetic reagent (*D. melanogaster*) | *UAS-Kir* | N/A | PMID:11222642 | |
| Genetic reagent (*D. melanogaster*) | *UAS-CsChrimson* | Bloomington *Drosophila* Stock Center | BDSC:82181; RRID:BDSC_82181 | |
| Genetic reagent (*D. melanogaster*) | *TrpA1$^{SH}$-Gal4* | Dr. Paul A. Garrity | PMID:18548007 | |
| Genetic reagent (*D. melanogaster*) | *R11F02-Gal4* | Bloomington *Drosophila* Stock Center | BDSC:49828; RRID:BDSC_49828 | |
| Genetic reagent (*D. melanogaster*) | *tubGal80$^{ts}$* | Bloomington *Drosophila* Stock Center | BDSC:7019; RRID:BDSC_7019 | |
| Genetic reagent (*D. melanogaster*) | *Orco$^1$* | Bloomington *Drosophila* Stock Center | BDSC:23129; RIDD:BDSC_23129 | |
| Genetic reagent (*D. melanogaster*) | *NPF$^{-/-}$* | Bloomington *Drosophila* Stock Center | BDSC:83722; RRID:BDSC_83722 | |
| Genetic reagent (*D. melanogaster*) | sNPF hypomorph | Bloomington *Drosophila* Stock Center | BDSC:85000; RRID:BDSC_85000 | |
| Genetic reagent (*D. melanogaster*) | Dh44-Gal4 | Dr. Greg Suh | PMID:21709242 | |
| Genetic reagent (*D. melanogaster*) | Akh-Gal4 on II | Bloomington *Drosophila* Stock Center | BDSC:25683; RRID:BDSC_25683 | |
| Genetic reagent (*D. melanogaster*) | Ilp6 LOF | Bloomington *Drosophila* Stock Center | BDSC:30885; RRID:BDSC_30885 | |
| Genetic reagent (*D. melanogaster*) | Unpaird3Δ | Bloomington *Drosophila* Stock Center | BDSC:55728; RRID:BDSC_55728 | |
| Genetic reagent (*D. melanogaster*) | Unpaird2Δ | Bloomington *Drosophila* Stock Center | BDSC:55727; RRID:BDSC_55727 | |
| Genetic reagent (*D. melanogaster*) | *period$^{01}$* | Kindly shared from Dr. Paul H Taghert | PMID:5002428 PMID:6435882 | |
| Genetic reagent (*D. melanogaster*) | *timeless$^{01}$* | Kindly shared from Dr. Patrick Emery | PMID:8128246 | |
| Chemical compound, drug | PTFE | Sigma-Aldrich | Cat# 665800 | |
| Chemical compound, drug | PTFE plus | byFormica | https://byformica.com/collections/shop-intl/products/ptfe-3pc | |
| Chemical compound, drug | DMSO | Sigma-Aldrich | #472301 | |
| Chemical compound, drug | ATR | Sigma-Aldrich | #R2500 | |
| Chemical compound, drug | Sucralose | Sigma-Aldrich | #69293 | |
| Chemical compound, drug | Glucose | Sigma-Aldrich | #G7021 | |
| Chemical compound, drug | Fructose | Sigma-Aldrich | #F3510 | |
| Software, algorithm | GraphPad Prism | Dotmatics | https://www.graphpad.com/features | |
| Software, algorithm | Microsoft Excel | Microsoft | https://www.microsoft.com/en-us/microsoft-365/excel | |
| Software, algorithm | FLIC R code master | Dr. David Fletcher lab, **PletcherLab, 2024** | https://github.com/PletcherLab/FLIC_R_Code | |

All flies were reared under 12-hr light/12-hr dark cycles at 25°C and 60–70% humidity in an incubator (DRoS33SD, Powers Scientific Inc) with an electric timer (light on: 8 am; light off: 8 pm). The light intensity was 1000–1400 lux. All flies were reared on custom fly food recipe, with the following composition per 1 l of food: 6.0 g sucrose, 7.3 g agar, 44.6 g cornmeal, 22.3 g yeast, and 16.3 ml molasses, as previously described (*Umezaki et al., 2018*). *white$^{1118}$* (*w$^{1118}$*) and *yellow$^1$ white$^1$* (*y$^1$w$^1$*)

flies were used as control flies. *EP5Δ; Gr64a¹ (Gr5a⁻/⁻; Gr64a⁻/⁻)* was kindly provided by Dr. Anupama Dahanukar (**Dahanukar et al., 2007**). *R1; Gr5a-LexA; +; ΔGr61a, ΔGr64a-f (Gr5a⁻/⁻; Gr61a⁻/⁻, Gr64a-f⁻/⁻)* were kindly provided by Dr. Hubert Amrein (**Yavuz et al., 2014**; **Fujii et al., 2015**). *Dh44-Gal4* was kindly provided by Dr. Greg Suh (**Dus et al., 2011**). *TrpA1^{SH}-gal4* was kindly provided by Dr. Paul A. Garrity (**Hamada et al., 2008**). Other fly lines were provided by Bloomington *Drosophila* stock center and Vienna *Drosophila* Stock Center.

## Temperature preference behavioral assay

The temperature preference behavior assays were examined using a temperatures gradient, set from 16 to 34°C and were performed for 30 min in an environmental room maintained at 25°C /60–70% humidity, as previously described (**Umezaki et al., 2018**). Because starved flies showed a lower preferred temperature (Tp), the temperature regulation was lower than usual (**Umezaki et al., 2018**, Current Biology). We prepared a total of 40–50 flies (male and female flies were mixed) for fed condition experiments and 90–100 flies for overnight(s) starved and refed condition experiments for one trial. Flies were never reused in subsequent trials. In the starved and refed experiments, we prepared twice the number of flies needed for a trial because almost half of them died from starvation stress. Others climbed on the wall and ceiling starved flies are usually hyperactive (**Yang et al., 2015**), even though we applied slippery coating chemicals (PTFE; Cat# 665800, Sigma or byFormica PTFE Plus, https://byformica.com/products/fluon-plus-ptfe-escape-prevention-coating) to the Plexiglass covers.

Behavioral assays were performed for 30 min at ZT4–7 (light on and light off are defined as ZT0 and ZT12, respectively), and starvation was initiated at ZT9–10 (starved for 1, 2, or 3 O/N are 18–21, 42–45, or 66–69 hr, respectively). As for STV1.5 for *Figure 3—figure supplement 1*, starvation was initiated at ZT1–2 (26–29 hr). For the starvation assays, the collected flies were maintained on our fly food for at least 1 day and then transferred to plastic vials containing 3 ml of distilled water, which was absorbed by a Kimwipe paper. For refeeding experiments, starved flies were transferred to plastic vials containing 2 ml of 2.8 mM sugar solution (sucralose water/glucose water). Sugar solution is absorbed by half the size of a Kimwipe. The details of the starvation period are described in the following section (see 'Starvation Condition').

After the 30-min behavioral assay, the number of flies whose bodies were completely located on the aluminum plate was counted. Flies whose bodies were partially or completely located on the walls of the Plexiglass cover were not included in the data analysis. The percentage of flies within each one-degree temperature interval on the apparatus was calculated by dividing the number of flies within each one-degree interval by the total number of flies on the apparatus. The location of each one-degree interval was determined by measuring the temperature at six different points on the bottom of the apparatus. Data points were plotted as the percentage of flies within a one-degree temperature interval. The weighted mean of Tp was calculated by summing the product of the percentage of flies within a one-degree temperature interval and the corresponding temperature (e.g., fractional number of flies × 17.5°C + fractional number of flies × 18.5°C +……… fractional number of flies × 32.5°C). Each experiment was performed with trials >5. If the SEM of the averaged Tp was not <0.3 after the five trials, additional trials were performed approximately 10 times until the SEM was <0.3.

Microsoft Excel (Home tab > Conditional formatting tool > 3-color scales and data bars) was used to create heat maps to show the distribution of flies in each experimental condition. The averaged percentages of flies that settled on the apparatus within each one-degree temperature interval were used to create the heat maps. Each scale value is as follows; minimum value: 0, midpoint value: 15%, and maximum value: 60% for *w^{1118}*. Minimum value: 0, middle value: 10%, and maximum value: 45% for Gr64fGal4>CsChrimson.

## Starvation conditions and recovery

Most of the flies were starved for two overnights (O/N). Because some flies (e.g., *ilp6* mutants) show starvation resistance and seem to be still healthy even after 2 O/N of starvation. We had to starve them for 3 O/N to show a significant difference in Tp between fed and starved flies. On the other hand, some flies (e.g., *w^{1118}* flies) are very sick after 3 O/N of starvation, in which case we only had to starve them for 1 day. Therefore, the starvation conditions we used for this manuscript are from 1 to 3 O/N.

First, we confirmed the starvation period by focusing on Tp which resulted in a statistically significant Tp difference between fed and starved flies; as mentioned above, flies prefer lower temperatures when starvation is prolonged (*Umezaki et al., 2018*). Therefore, when Tp was not statistically different between fed and starved flies, we extended the starvation period from 1 to 3 O/N. Importantly, we shown in *Figure 3—figure supplement 1* that the duration of starvation does not affect the recovery effect. Furthermore, $w^{1118}$ flies cannot survive 42–49 or 66–69 hr of starvation.

## TPR assay

For the TPR assays, we performed temperature preference behavior assays in different time windows during the daytime (ZT or circadian time (CT) 1–3, 4–6, 7–9, and 10–12) as described previously (*Kaneko et al., 2012*; *Goda et al., 2014*). Because starvation duration directly affects flies' Tp (*Umezaki et al., 2018*) starvation was initiated at each time window to adjust the starvation duration at each time point, which means flies were starved for 24 or 48 hr but not 1 or 2 O/N. Each behavioral assay was not examined during these time periods (ZT or CT0–1 and 11.5–12) because of large phenotype variation around light on and light off.

Furthermore, insulin levels were shown to peak at 10 min and gradually decline (*Tsao et al., 2023*). Also, how quickly the flies can consume food is unclear. These factors may influence temperature preference behavior. Therefore, to minimize these effects, we decided to test the temperature preference behavioral assay immediately after the flies had eaten the food.

## Optogenetic activation

For the optogenetic activation of the target neurons for behavioral assays, the red-light-sensitive channelrodopsin, *UAS-CsChrimson*, was crossed with each Gal4 driver. Flies were reared on fly food at 25°C and 60–70% humidity under LD cycles in an incubator (DRoS33SD, Powers Scientific Inc) with an electric timer. After the flies emerged, adult flies were collected and maintained on fly food for 1–2 days. The next day, flies were transferred to water with or without 0.8 mM ATR (#R2500, Sigma) diluted in dimethyl sulfoxide (#472301, Sigma) for 2 O/N. To activate flies expressing UAS-CsChrimson crossed with Gal4 drivers, we used a 627 nm red light-emitting diode equipped with a pulsed photoillumination system (10 Hz, 0.08 mW mm$^{-2}$). Flies were exposed to pulsed red light for 10 min, which corresponds to the refeeding period. This photoillumination system was used in an incubator (Sanyo Scientific, MIR-154) and followed by temperature preference behavioral assays.

The *R11F02-Gal4>uas-CsChrimson* flies do not develop into adults and die in the pupal stage. Therefore, the Gal4/Gal80ts system was used to restrict *uas-CsChrimson* expression. The *Gal80$^{ts}$* is a temperature-sensitive allele of *Gal80* that causes *Gal4* inhibition at 18°C and activation at 29°C (*Southall and Brand, 2008*).The *tubGal80$^{ts}$; R11F02-Gal4>uas-CsChrimson* flies were reared on fly food at 18°C. Emerging adult flies were collected and kept on fly food at 29°C. The next day, flies were transferred to water (starved condition) with or without 0.8 mM ATR for 2 O/N. Starved flies with or without ATR application were exposed to pulsed red light for 10 min (equivalent to the refeeding period) and then immediately loaded into the behavioral apparatus for behavioral assays to measure their Tp. All flies were treated with ATR after they had fully developed into the adults. This means that Gal4-expressing cells were activated by red light via CsChrimson only at adult stages.

## Feeding assay

To measure individual fly feeding, we used the FLIC system (*Ro et al., 2014*). Groups of 1- to 2-day-old male and female flies were starved for 24 hr starting between ZT4 and ZT5 (12–1 pm). Individual flies were then loaded into FLIC monitors. Flies were acclimated to the monitors for 30 min with access to water in the feeding wells. At the start of the feeding study, water was replaced with 2.8 mM sucralose or 2.8 mM glucose solution (equivalent to 5% glucose concentration) and the number of licks (touches) was recorded for 30 min. Water, sucralose, or glucose water was administered individually in separate experiments. Assays were performed on ZT4–7. To account for the potential confounding effect of startle response when food is changed, wells where water was replaced with new water were used as a control. FLIC raw data were analyzed using the FLIC R code master (Pletcher Lab, copy archived at *PletcherLab, 2024*). Lick counts were obtained and summed in 5-min window bins, while cumulative licks were obtained by successively summing the licks in these bins.

## Acknowledgements

We are grateful to Drs. Anupama Dahanukar, Hubert Amrein, Greg Suh, and Paul A Garrity and the Bloomington *Drosophila* Fly Stock Center for the fly lines. We thank members of the Hamada laboratory for critical comments and advice on the manuscript, Dr. Richard A Lang and members of his laboratory for comments and kindly sharing reagents, Dr. Satoshi Namekawa for kindly sharing reagents, and Matthew Batie and Nathan T Petts for design and construction of the behavioral assays and red-light illumination apparatus. This research was supported by a RIP funding from Cincinnati Children's Hospital, JST (Japan Science and Technology)/Precursory Research for Embryonic Science and Technology (PRESTO), the March of Dimes, and NIH R01 grant GM107582, NIH R21 grant NS112890, NIH R35 GM152154 grant, and NIH R34 grant NS132843 to FNH. SHS is a Latin American Fellow in the Biomedical Sciences supported by the Pew Charitable Trusts and by NIH K99 grant NS133470. Research in the laboratory of JCC is supported by NIH R01 grant DK124068.

## Additional information

### Funding

| Funder | Grant reference number | Author |
|---|---|---|
| Precursory Research for Embryonic Science and Technology | | Fumika Hamada |
| March of Dimes Foundation | | Fumika Hamada |
| National Institute of General Medical Sciences | R01GM107582 | Fumika Hamada |
| National Institute of Neurological Disorders and Stroke | R21NS112890 | Fumika Hamada |
| National Institute of General Medical Sciences | R35GM152154 | Fumika Hamada |
| National Institute of Neurological Disorders and Stroke | R34NS132843 | Fumika Hamada |
| Pew Charitable Trusts | | Sergio Hidalgo |
| National Institute of Neurological Disorders and Stroke | K99NS133470 | Sergio Hidalgo |
| National Institute of Diabetes and Digestive and Kidney Diseases | R01DK124068 | Joanna Chiu |

The funders had no role in study design, data collection, and interpretation, or the decision to submit the work for publication.

### Author contributions

Yujiro Umezaki, Conceptualization, Data curation, Formal analysis, Investigation, Methodology, Project administration, Writing – review and editing; Sergio Hidalgo, Data curation, Formal analysis, Funding acquisition, Writing – review and editing; Erika Nguyen, Tiffany Nguyen, Jay Suh, Sheena S Uchino, Data curation, Formal analysis; Joanna Chiu, Supervision, Funding acquisition, Investigation, Writing – review and editing; Fumika Hamada, Conceptualization, Formal analysis, Supervision, Funding acquisition, Validation, Investigation, Writing - original draft, Project administration, Writing – review and editing

### Author ORCIDs

Yujiro Umezaki ⓘ https://orcid.org/0009-0008-1136-8464

Sergio Hidalgo ⓘ https://orcid.org/0000-0002-2604-156X
Joanna Chiu ⓘ https://orcid.org/0000-0001-7613-8127
Fumika Hamada ⓘ https://orcid.org/0009-0003-6537-7386

Reviewer #1 (Public Review): https://doi.org/10.7554/eLife.94703.3.sa1
Reviewer #2 (Public Review): https://doi.org/10.7554/eLife.94703.3.sa2
Author response https://doi.org/10.7554/eLife.94703.3.sa3

---

## Additional files

### Supplementary files

• MDAR checklist

### Data availability

All data generated or analyzed during this study are included in the manuscript and supporting files; source data files have been provided.

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
