## [Editor Report · eLife assessment]

This paper presents **valuable** findings that gustation and nutrition might independently influence the preferred environmental temperature in flies. The evidence supporting the main claims is **solid** and well presented. The finding that flies might thus exhibit a cephalic phase response similar to mammals will be of value for future investigations.

---

## [Referee Report · Reviewer #1 (Public Review)]

Summary:

This paper presents valuable findings that gustation and feeding state influence the preferred environmental temperature preference in flies. Interestingly, the authors showed that by refeeding starved animals with non-nutritive sugar sucralose, they are able to tune their preference towards a higher temperature in addition to nutrient-dependent warm preference. The authors show that temperature sensing and sweet sensing gustatory neurons (SGNs) are involved in the former but not the latter. In addition, their data indicate that peptidergic signals involved in internal state and clock genes are required for taste-dependent warm preference behavior.

The authors made an analogy of their results to the cephalic phase response (CPR) in mammals where the thought, sight and taste of food prepares the animal for the consumption of food and nutrients. The authors showed that taste triggers CPR-induced temperature preference behaviors in flies. The authors also briefly covered that the combined modalities of smell and taste induced CPR responses, showing that starved orco mutant flies failed to recover temperature preference after refeeding with sucralose.

The findings of this work hold promising future research prospects, for example, whether the sight of food influences temperature preference behavior in hungry flies, or whether taste, smell and sight work together or independently in promoting CPR responses.

Futhermore, these valuable behavioral results can be further investigated in flies with the advantage of being able to dissect the neural circuitry underlying CPR and nutrient homeostasis.

Strengths:

(1) The authors convincingly showed that tasting is sufficient to drive warm temperature preference behavior in starved flies and show that it is independent of nutrient-driven warm preference.

(2) By using the genetic manipulation of key internal sensors and genes controlling internal feeding and sleep state such as DH44 neurons and the per genes for eg the authors linked gustation and temperature preference behavior control to the internal state of the animal.

Weaknesses:

Most of the weaknesses of the paper have been addressed in the revision. The points mentioned below are meant to improve readability of the paper and to promote understanding of the significance of the work.

(1) Supplementary fig 1 could replace Figure 1A. The purpose of Figure 1F is not clear to me as the comparison between the different food substances is not separately addressed anywhere in the text.

(2) The data for the orco receptor mutant could be placed in the main figures to justify the discussion emphasising CPR-like responses.

---

## [Referee Report · Reviewer #2 (Public Review)]

Animals constantly adjust behavior and physiology based on internal states. Hungry animals, desperate for food, exhibit physiological changes immediately upon sensing, smelling, or chewing food, known as the cephalic phase response (CPR), involving processes like increased saliva and gastrointestinal secretions. While starvation lowers body temperature, the mechanisms underlying how the sensation of food without nutrients induces behavioral responses remain unclear. Hunger stress induces changes in both behavior and physiological responses, which in flies (or at least in *Drosophila melanogaster*) leads to a preference for lower temperatures, analogous to the hunger-driven lower body temperature observed in mammals. In this manuscript, the authors have used *Drosophila melanogaster* to investigate the issue of whether taste cues can robustly trigger behavioral recovery of temperature preference in starving animals. The authors find that food detection triggers a warm preference in flies. Starved flies recover their temperature preference after food intake, with a distinction between partial and full recovery based on the duration of refeeding. Sucralose, an artificial sweetener, induces a warm preference, suggesting the importance of food-sensing cues. The paper compares the effects of sucralose and glucose refeeding, indicating that both taste cues and nutrients contribute to temperature preference recovery. The authors show that that sweet gustatory receptors (Grs) and sweet GRNs (Gustatory Receptor Neurons) play a crucial role in taste-evoked warm preference. Optogenetic experiments with CsChrimson support the idea that the excitation of sweet GRNs leads to a warm preference. The authors then examine the internal state's influence on taste-evoked warm preference, focusing on neuropeptide F (NPF) and small neuropeptide F (sNPF), analogous to mammalian neuropeptide Y. Mutations in NPF and sNPF result in a failure to exhibit taste-evoked warm preference, emphasizing their role in this process. However, these neuropeptides appear not to be critical for nutrient-induced warm preference, as indicated by increased temperature preference during glucose and fly food refeeding in mutant flies. The authors also explore the role of hunger-related factors in regulating taste-evoked warm preference. Hunger signals, including diuretic hormone (DH44) and adipokinetic hormone (AKH) neurons, are found to be essential for taste-evoked warm preference but not for nutrient-induced warm preference. Additionally, insulin-like peptide 6 (Ilp6) and Unpaired3 (Upd3), related to nutritional stress, are identified as crucial for taste-evoked warm preference. The investigation then extends into circadian rhythms, revealing that taste-evoked warm preference does not align with the feeding rhythm. While flies exhibit a rhythmic feeding pattern, taste-evoked warm preference occurs consistently, suggesting a lack of parallel coordination. Clock genes, crucial for circadian rhythms, are found to be necessary for taste-evoked warm preference but not for nutrient-induced warm preference.

Strengths:

A well-written and interesting study, investigating an intriguing issue. The claims, none of which to the best of my knowledge controversial, are backed by a substantial number of experiments.

Weakness:

The experimental setup used and the procedures for assessing the temperature preferences of flies is rather sparingly described. Additional details and data presentation would enhance the clarity and replicability of the study. I kindly request the authors to consider the following points: (i) A schematic drawing or diagram illustrating the experimental setup for the temperature preference assay would greatly aid readers in understanding the spatial arrangement of the apparatus, temperature points, and the positioning of flies during the assay. The drawing should also be accompanied by specific details about the setup (dimensions, material, etc). (ii) It would be beneficial to include a visual representation of the distribution of flies within the temperature gradient on the apparatus. A graphical representation, such as a heatmaps or histograms, showing the percentage of flies within each one-degree temperature bin, would offer insights into the preferences and behaviors of the flies during the assay. In addition to the detailed description of the assay and data analysis, the inclusion of actual data plots, especially for key findings or representative trials, would provide readers with a more direct visualization of the experimental outcomes. These additions will not only enhance the clarity of the presented information but also provide the reader with a more comprehensive understanding of the experimental setup and results. I appreciate the authors' attention to these points and look forward to the potential inclusion of these elements in the revised manuscript.

Update: The revised manuscript now includes heatmaps showing the distribution of the flies across the temperature bins. As well as a schematic drawing of the behavioral setup.

---

## [Author Response]

The following is the authors’ response to the original reviews.

Main points:

(1) We have added data for fructose in Fig. 1

(2) We have added sta1s1cs (red stars and NS) comparing Tp between fed and refed flies.

(3) We have modified the figure for each point to the opened small circles.

(4) We have moved the data from Fig. S3 to Fig. 2 and 3.

(5) We have added the schema1c diagrams depic1ng behavioral assay in Fig. S1.

(6) We have added heatmaps for WT and Gr64f-Gal4>UAS-CsChrimson flies in Fig. S2.

(7) We have added Orco1 mutant data in Fig. S4.

**Public Reviews:**

**Reviewer #1 (Public Review):**
Summary:This paper presents valuable findings that gustation and feeding state influence the preferred environmental temperature preference in flies. Interestingly, the authors showed that by refeeding starved animals with the non-nutritive sugar sucralose, they are able to tune their preference towards a higher temperature in addition to nutrient-dependent warm preference. The authors show that temperature-sensing and sweet-sensing gustatory neurons (SGNs) are involved in the former but not the latter. In addition, their data indicate that pep3dergic signals involved in internal state and clock genes are required for taste-dependent warm preference behavior.The authors made an analogy of their results to the cephalic phase response (CPR) in mammals where the thought, sight, and taste of food prepare the animal for the consumption of food and nutrients. They further linked this behavior to core regulatory genes and peptides controlling hunger and sleep in flies having homologues in mammals. These valuable behavioral results can be further inves3gated in flies with the advantage of being able to dissect the neural circuitry underlying CPR and nutrient homeostasis.Strengths:(1) The authors convincingly showed that tasting is sufficient to drive warm temperature preference behavior in starved flies and that it is independent of nutrient-driven warm preference.(2) By using the genetic manipulation of key internal sensors and genes controlling internal feeding and sleep states such as DH44 neurons and the per genes for example, the authors linked gustation and temperature preference behavior control to the internal state of the animal.Weaknesses:(1) The title is somewhat misleading, as the term homeostatic temperature control linked to gustation only applies to starved flies.

We agree with the reviewer's suggestion and have changed the title to "Taste triggers a homeostatic temperature control in hungry flies".

(2) The authors used a temperature preference assay and refeeding for 5 minutes, 10 minutes, and 1 hour.Experimentally, it makes a difference if the flies are tested immediately after 10 minutes or at the same 3me point as flies allowed to feed for 1 hour. Is 10 minutes enough to change the internal state in a nutrition-dependent manner? Some of the authors' data hint at it (e.g. refeeding with fly food for 10 minutes), but it might be relevant to feed for 5/10 minutes and wait for 55/50min to do the assays at comparable time points.

Thank you for your suggestions. The temperature preference behavioral test itself takes 30 minutes from the time the flies are placed in the apparatus until the final choice is made. This means that after the hungry flies have been refed for 5 minutes, they will determine their preferred temperature within 35 minutes. It has been shown that insulin levels peak at 10 minutes and gradually decline (Tsao, et al., PLoS Genetics 2023). However, it is unclear how subtle insulin levels affect behavior and how quickly the flies are able to consume food. These factors may contribute to temperature preference in flies. Therefore, to minimize "extraneous" effects, we decided to test the behavioral assay immediately after they had eaten the food. We have noted in the material and method section that why we chose the condition based on behavior duration and insulin effect.

(3) A figure depicting the temperature preference assay in Figure 1 would help illustrate the experimental approach. It is also not clear why Figure 1E is shown instead of full statistics on the individual panels shown above (the data is the same).

We have revised Figure 1A and added statistics in Figure 1BCD. We also added a figure depicting the temperature preference assay (Fig. S1).

(4) The authors state that feeding rate and amount were not changed with sucralose and glucose. However, the FLIC assay they employed does not measure consumption, so this statement is not correct, and it is unclear if the intake of sucralose and glucose is indeed comparable. This limits some of the conclusions.

We agree and removed “amount” and have revised the MS.

(5) The authors make a distinction between taste-induced and nutrient-induced warm preference. Yet the statistics in most figures only show the significance between the starved and refed flies, not the fed controls. As the recovery is in many cases incomplete and used as a distinction of nutritive vs nonnutritive signals (see Figure 1E) it will be important to also show these additional statistics to allow conclusions about how complete the recovery is.

We agree with the comments and have revised the MS and figures.

(6) The starvation period used is ranging from 1 to 3 days, as in some cases no effect was seen upon 1 day of starvation (e.g. with clock genes or temperature sensing neurons). While the authors do provide a comparison between 18-21 and 26-29 hours old flies in Figure S1, a comparison for 42-49 and 66-69 hours of starvation is missing. This also limits the conclusion as the "state" of the animal is likely quite different after 1 day vs. 3 days of starvation and, as stated by the authors, many flies die under these conditions.

We mainly used 2 overnights of starvation. Some flies (e.g. Ilp6 mutants) were completely healthy even after 2 overnights of starvation, we had to starve them for 3 overnights. For example, Ilp6 mutants needed 3 overnights of starvation to show a significant difference Tp between fed and starved flies. On the other hand, some flies (e.g. w1118 control flies) were very sick after 2 overnights of starvation, we had to starve them for one overnight. Therefore, the starvation conditions which we used for this manuscript are from 1- 3-overnights.

First, we confirmed the starvation time by focusing on Tp which resulted in a sta1s1cally significant Tp difference between fed and starved flies; as men1oned above, flies prefer lower temperatures when starvation is prolonged (Umezaki et al., Current Biology 2018). Therefore, if Tp was not statistically different between fed and starved flies, we extended the starva1on 1me from 1 to 3 overnights. Importantly, we show in Fig. S3 that the dura1on of starvation did not affect the recovery effect. Furthermore, since control flies do not survive 42-49 or 66-69 hours of starvation, we can not test the reviewer's suggestion. We have carefully documented the conditions in the Material and method and figure legends.

(7) In Figure 2, glucose-induced refeeding was not tested in Gr mutants or silenced animals, which would hint at post-ingestive recovery mechanisms related to nutritional intake. This is only shown later (in Figure S3) but I think it would be more fitting to address this point here. The data presented in Figure S3 regarding the taste-evoked vs nutrient-dependent warm preference is quite important while in some parts preliminary. It would nonetheless be justified to put this data in the main figures. However, some of the conclusions here are not fully supported, in part due to different and low n numbers, which due to the inherent variability of the behavior do not allow statistically sound conclusions. The authors claim that sweet GRNs are only involved in taste-induced warm preference, however, glucose is also nutritive but, in several cases, does not rescue warm preference at all upon removal of GRN function (see Figures S3A-C). This indicates that the Gal4 lines and also the involved GRs are potentially expressed in tissues/neurons required for internal nutrient sensing.

Thank you for your suggestion. We have added Figure S3ABC (glucose refeeding using Gr mutants and silenced animals) to Figure 2. There is no low N number since we tested > 5 times, i.e. >100 flies were tested. Tp may have a variation probably due to the effect of starvation on their temperature preference.

We did not mention that "The authors claim that sweet GRNs are only involved in taste-induced warm preference...". However, our wri1ng may not be clear enough. We agree that "...GRs may be expressed in tissues/neurons required for internal nutrient sensing. ..." We have rewritten and revised the section.

(8) In Figure 4, fly food and glucose refeeding do not fully recover temperature preference after refeeding. With the statistical comparison to the fed control missing, this result is not consistent with the statement made in line 252. I feel this is an important point to distinguish between state-dependent and taste/nutrition-dependent changes.

We inserted the statistics and compared between Fed and other conditions.

(9) The conclusion that clock genes are required for taste-evoked warm preference is limited by the observation that they ingest less sucralose. In addition, the FLIC assay does not allow conclusions about the feeding amount, only the number of food interactions. Therefore, I think these results do not allow clear-cut conclusions about the impact of clock genes in this assay.

We agree and remove “amount” and have revised the MS. The per01 mutants ate (touched) sucralose more often than glucose. On the other hand, 1m01 mutants ate glucose more often than sucralose (Figure S6BC). However, these mutants s1ll showed a similar TP pattern for sucralose and glucose refeeding (Fig. 5CD). The results suggest that the 1m01 flies eat enough amount of sucralose over glucose that their food intake does not affect the TP behavioral phenotype. We have rewritten and revised the section.

(10) CPR is known to be influenced by taste, thought, smell, and sight of food. As the discussion focused extensively on the CPR link to flies it would be interesting to find out whether the smell and sight of food also influence temperature preference behavior in animals with different feeding states.

We have added the data using Olfactory receptor co-receptor (Orco1) mutant, which lack olfaction, in Fig. S4. They failed to show the taste-evoked warm preference, but exhibited the nutrient-induced warm preference. Therefore, the data suggest that olfactory detection is also involved in taste-evoked warm preference. On the other hand, "seeing food" is probably more complicated, since light dramatically affects temperature preference behavior and the circadian clock that regulates temperature preference rhythms. Therefore, it will not be unlikely to draw a solid conclusion from the short set of experiments. We will address this issue in the next study.

(11) In the discussion in line 410ff the authors claim that "internal state is more likely to be associated with taste-evoked warm preference than nutrient-induced warm preference." This statement is not clear to me, as neuropeptides are involved in mediating internal state signals, both in the brain itself as well as from gut to brain. Thus, neuropeptidergic signals are also involved in nutrient-dependent state changes, the authors might just not have identified the peptides involved here. The global and developmental removal of these signals also limits the conclusions that can be drawn from the experiments, as many of these signals affect different states, circuits, and developmental progression.

We agree with the comments. We have removed the sentences and revised the MS.

**Reviewer #2 (Public Review):**
Animals constantly adjust their behavior and physiology based on internal states. Hungry animals, desperate for food, exhibit physiological changes immediately upon sensing, smelling, or chewing food, known as the cephalic phase response (CPR), involving processes like increased saliva and gastrointestinal secretions. While starvation lowers body temperature, the mechanisms underlying how the sensation of food without nutrients induces behavioral responses remain unclear. Hunger stress induces changes in both behavior and physiological responses, which in flies (or at least in *Drosophila melanogaster*) leads to a preference for lower temperatures, analogous to the hunger-driven lower body temperature observed in mammals. In this manuscript, the authors have used *Drosophila melanogaster* to investigate the issue of whether taste cues can robustly trigger behavioral recovery of temperature preference in starving animals. The authors find that food detection triggers a warm preference in flies. Starved flies recover their temperature preference after food intake, with a distinction between partial and full recovery based on the duration of refeeding. Sucralose, an artificial sweetener, induces a warm preference, suggesting the importance of food-sensing cues. The paper compares the effects of sucralose and glucose refeeding, indicating that both taste cues and nutrients contribute to temperature preference recovery. The authors show that sweet gustatory receptors (Grs) and sweet GRNs (Gustatory Receptor Neurons) play a crucial role in taste-evoked warm preference. Optogenetic experiments with CsChrimson support the idea that the excitation of sweet GRNs leads to a warm preference. The authors then examine the internal state's influence on taste-evoked warm preference, focusing on neuropeptide F (NPF) and small neuropeptide F (sNPF), analogous to mammalian neuropeptide Y. Mutations in NPF and sNPF result in a failure to exhibit taste-evoked warm preference, emphasizing their role in this process. However, these neuropeptides appear not to be critical for nutrient-induced warm preference, as indicated by increased temperature preference during glucose and fly food refeeding in mutant flies. The authors also explore the role of hunger-related factors in regula3ng taste-evoked warm preference. Hunger signals, including diuretic hormone (DH44) and adipokinetic hormone (AKH) neurons, are found to be essential for taste-evoked warm preference but not for nutrient-induced warm preference. Additionally, insulin-like peptides 6 (Ilp6) and Unpaired3 (Upd3), related to nutritional stress, are identified as crucial for taste-evoked warm preference. The investigation then extends into circadian rhythms, revealing that taste-evoked warm preference does not align with the feeding rhythm. While flies exhibit a rhythmic feeding pattern, taste-evoked warm preference occurs consistently, suggesting a lack of parallel coordination. Clock genes, crucial for circadian rhythms, are found to be necessary for taste-evoked warm preference but not for nutrient-induced warm preference.Strengths:A well-written and interesting study, investigating an intriguing issue. The claims, none of which to the best of my knowledge controversial, are backed by a substantial number of experiments.Weakness:The experimental setup used and the procedures for assessing the temperature preferences of flies are rather sparingly described. Additional details and data presentation would enhance the clarity and replicability of the study. I kindly request the authors to consider the following points:i) A schematic drawing or diagram illustrating the experimental setup for the temperature preference assay would greatly aid readers in understanding the spatial arrangement of the apparatus, temperature points, and the positioning of flies during the assay. The drawing should also be accompanied by specific details about the setup (dimensions, material, etc).

Thank you for your suggestions. We have added the schematic drawing in Fig. S1.

ii) It would be beneficial to include a visual representation of the distribution of flies within the temperature gradient on the apparatus. A graphical representation, such as a heatmaps or histograms, showing the percentage of flies within each one-degree temperature bin, would offer insights into the preferences and behaviors of the flies during the assay. In addition to the detailed description of the assay and data analysis, the inclusion of actual data plots, especially for key findings or representative trials, would provide readers with a more direct visualization of the experimental outcomes. These additions will not only enhance the clarity of the presented information but also provide the reader with a more comprehensive understanding of the experimental setup and results. I appreciate the authors' attention to these points and look forward to the potential inclusion of these elements in the revised manuscript.

Thank you for the advice. We have added the heat map for WT and Gr64fGal4>CsChrimson data in Fig. S2.

**Reviewer #3 (Public Review):**
Summary:The manuscript by Yujiro Umezaki and colleagues aims to describe how taste stimuli influence temperature preference in *Drosophila*. Under starvation flies display a strong preference for cooler temperatures than under fed conditions that can be reversed by refeeding, demonstrating the strong impact of metabolism on temperature preference. In their present study, Umezaki and colleagues observed that such changes in temperature preference are not solely triggered by the metabolic state of the animal but that gustatory circuits and peptidergic signalling play a pivotal role in gustation-evoked alteration in temperature preference.The study of Umezaki is definitively interesting and the findings in this manuscript will be of interest to a broad readership.Strengths:The authors demonstrate interesting new data on how taste input can influence temperature preference during starvation. They propose how gustatory pathways may work together with thermosensitive neurons, peptidergic neurons and finally try to bridge the gap between these neurons and clock genes. The study is very interesting and the data for each experiment alone are very convincing.Weaknesses:In my opinion, the authors have opened many new questions but did not fully answer the initial question - how do taste-sensing neurons influence temperature preferences? What are the mechanisms underlying this observation? Instead of jumping from gustatory neurons to thermosensitive neurons to peptidergic neurons to clock genes, the authors should have stayed within the one question they were asking at the beginning. How does sugar sensing influence the physiology of thermos-sensation in order to change temperature preference? Before addressing all the following question of the manuscript the authors should first directly decipher the neuronal interplay between these two types of neurons.
**Recommendations for the authors:**

**Reviewer #1 (Recommendations For The Authors):**
Figure S3D is cited before S2, so please rearrange the numbering.

Thank you. We have changed the numbering.

I would also suggest a different color to visualize the data points in Figure S3, as some are barely visible on the dark bars (e.g. on a dark green background).

We have revised the figures. The data points were changed to smaller opened circles.

**Reviewer #2 (Recommendations For The Authors):**
*Please, expand on the experimental procedure, and describe the assay in detail.

We have added a scheme for the assay in Fig. S1 and also have revised the manuscript and figures.

*Show the distribution of the gradient data that the preference values are based upon. Not necessarily for all, but for select key experiments. Heatmaps for each replicate (stacked on top of each other) would be a nice way of showing this. Simple histograms would of course work as well.

We have added heatmaps of selected key experiments that were added in Fig. S2. We have revised the manuscript and figures, correspondingly.

**Reviewer #3 (Recommendations For The Authors)**
The manuscript by Yujiro Umezaki and colleagues aims at describing how taste stimuli influence temperature preference in *Drosophila*. Under starvation, flies display a strong preference for cooler temperatures than under-fed conditions that can be reversed by refeeding, demonstrating the strong impact of metabolism on temperature preference. In their present study, Umezaki and colleagues observed that such changes in temperature preference are not solely triggered by the metabolic state of the animal but that gustatory circuits play a pivotal role in temperature preference. The study of Umezaki is definitively interesting and the findings in this manuscript will be of interest to a broad readership. However, I would like to draw the authors' attention to some points of concern:The title to me sounds somehow inadequate. The definition of homeostasis (Cambridge Dictionary) is as follows: "the ability or tendency of a living organism, cell, or group to keep the conditions INSIDE it the same despite any changes in the conditions around it, or this state of internal balance". What do the authors mean by homeostatic temperature control? Reading the title not knowing much about poikilotherm insects I would understand that the authors claim that Drosophila can indeed keep a temperature homeostasis as mammals do. As *Drosophila* is not a homoiotherm animal and thus cannot keep its body temperature stable the title should be amended.

Homeostasis means a state of balance between all the body systems necessary for the body to survive and function properly. *Drosophila* are ectotherms, so the source of temperature comes from the environment, and their body temperature is very similar to that of their environment. However, the flies' temperature regulation is not simply a passive response to temperature. Instead, they actively seek a temperature based on their internal state. We have shown that the preferred temperature increases during the day and decreases during the night, showing a circadian rhythm of temperature preference (TPR). Because their environmental temperature is very close to their body temperature, TPR gives rise to body temperature rhythms (BTR). We have shown that TPR is similar to BTR in mammals. (Kaneko et al., Current Biology 2012 and Goda et al., JBR 2023). Similarly, we showed that the hungry flies choose a lower temperature so that the body temperature is also lower. Therefore, our data suggest that the fly maintains its homeostasis by using the environmental temperature to adjust its body temperature to an appropriate temperature depending on its internal state. Therefore, I would like to keep the title as "Taste triggers a homeostatic temperature control in hungry flies" We have added more explana1on in the Introduc1on and Discussion.

Accordingly, the authors compare the preference of flies to cooler temperatures to the reduced body temperature of mammals (Lines 64 - 65). However, according to the cited literature the reduced body temperature in starved rats is discussed to reduce metabolic heat production (Sakurada et al., 2000). The authors should more rigorously give a short summary of the findings in the cited papers and the original interpretation to help the reader not get confused.

In flies, it has been shown that a lower temperature means a lower metabolic rate, and a higher temperature means a higher metabolic rate. Therefore, hungry flies choose a lower temperature where their metabolic rate is lower and they do not need as much heat.

Similarly, in mammals, starvation causes a lower body temperature, hypothermia. Body temperature is controlled by the balance between heat loss and heat production. The starved mammals showed lower heat production. We have added this information to the introduction.

The authors show that 5 min fly food refeeding causes a par3al recovery of the naïve temperature preference of the flies (Figure 1B) and that feeding of sucralose par3ally rescues the preference whereas glucose rescues the preference similar to refeeding with fly food would do. As glucose is both sweet and metabolically valuable it would be clearer for the reader if the authors start with the fly food experiment and then show the glucose experiment to show that the altered temperature preference depends on the food component glucose. From there they can further argue that glucose is both sweet (hedonic value) and metabolically valuable. And to disentangle sweetness from metabolism one needs a sugar that is sweet but cannot be metabolized - sucralose.

Thank you for your advice. Since the data with sucralose is the one we want to highlight the most, we decided to present it in the order of sucralose, glucose, and fly food.

In the sucralose experiment the authors omit the 5 min data point and only show the 10 min time point. As Figure 1F indicates that both Glucose and Sucralose elicit the same attractiveness in the flies and that sweetness influences the temperature preference, it is important that the authors show the 5 min temperature preference too to underline the effect of the sweet taste stimulus on the fly behavior independent from the caloric value. Further, the authors should demonstrate not only the cumulative touches but how much sucralose or glucose may already be consumed by the fly in the depicted time frames.

It is interesting to see how much sucralose or glucose the flies consume over the time frames shown. Although the cumula1ve exposure to sugar is ideally equivalent to the amount of sugar, we need a different way to actually measure the amount of sugar. We will now emphasize "cumulative touches" rather than "amount of sugar" in the text. In the next study, we will look at how much sucralose or glucose the fly has already consumed.

Sucralose and Glucose have a similar molecular structure - it would be interesting to see how the sweet taste of a sugar with a different molecular structure like fructose and its receptor Gr43b (Myamato & Amrein 2014) may contribute to temperature preferences.

Sucralose and Glucose are not structurally similar. That said, we tested fructose refeeding anyway. The hungry flies showed a taste-evoked warm preference after fructose refeeding. We have added data in Figure 1E and F. The data suggest that sweet taste is more important than sugar structure. We also tested Gr43b>CsChrimson. However, the flies do not show the taste-evoked warm preference (data not shown). The data suggest that Gr43b is not the major receptor controlling taste-evoked warm preference. We have revised the manuscript.

Both sugars appear similarly attractive to the flies (Figure 1F) - are water, sucralose, and glucose presented in a choice assay or are these individually in separate experiments?

Water, sucralose, and glucose were individually presented in separate experiments. We clarified it in the figure legend.

Subsequently, the authors address the question of how sweet taste may influence temperature preferences in flies. To this end, the authors first employ gustatory receptor mutants for Gr5a, Gr64a, and Gr61a and demonstrate that sucralose feeding does not rescue temperature preference in the absence of sweet taste receptors. In an alternative approach, the authors do not use mutants but an expression of UAS:Kir in Gr64F neurons. Taking a closer look at the graph it appears that the Kir expressing flies have an increased (nearly 1{degree sign}C) temperature preference than the starved mutant flies. Is this preference change related to the mutation directly and what would be the result if Kir would be conditionally only expressed after development is completed, or is the observed temperature preference related to the Gr64f-Gal4 line? If the latter would be the case perhaps the authors may want to bring the flies to the same genetic background to allow for a more direct comparison of the temperature preferences.

The Gr64fGal4>Kir flies show a ~one degree higher preferred temperature under starvation compared to the mutants. However, the phenotype is similar to the controls, Gr64fGal4/+ flies, under starvation. Therefore, this phenotype is not due to either the mutation or the Kir effect. Most importantly, the Gr64fGal4>Kir flies failed to show a taste-evoked warm preference. Together with other mutant data, we concluded that sweet GRNs are required for taste-evoked warm preference.

Overall, the figure legend for Figure 2 is very cryptic and should be more detailed.

We have revised the figure legend for Figure 2.

To shed light on the mechanisms underlying the changes in temperature preferences through gustatory stimuli the authors next blocked heat and cold sensing neurons in fed and starved flies and found out that TrpA1 expressing anterior cells and R11F02-Gal4 expressing neurons both participate in sweetness-induced alteration of temperature preference in starved animals. At this point, it should be explicitly indicated in the figure that the flies need more than one overnight starva3on to display the behavior (Figure 3A).

We have revised the manuscript.

The data provided by the authors indicate a kind of push-and-pull mechanism between heat and cold-sensing neurons under starvation that is somehow influenced by sweet taste sensing. Further, the authors demonstrate that TrpA1-as well as R11F02-Gal4 driven Chrimson activation is sufficient to partially rescue temperature preference under starvation. At this point is unclear why the authors use a tubGal80ts expression system but not for the TrpA1SH-Gal4 driven Chrimson. As the development itself and the conditions under which the animals were raised may have influence on the temperature preference it is important that both groups are equally raised if the authors want to directly compare with each other.

As we wrote in the Material and Method, the R11F02-Gal4>uas-CsChrimson flies died during the development. Therefore, we had to use tubGal80ts. On the other hand, the TrpA1-Gal4>CsChrimson flies can survive to adults. As we mentioned in MS, all flies were treated with ATR after they had fully developed into adults. This means that both TrpA1-Gal4 and R11F02-Gal4 expressing cells are ac1vated by red light via CsChrimson only in adult stages. We carefully revised the MS.

It is a pity that the authors at this point have decided to not deepen the understanding of the circuitry between thermo-sensation and metabolic homeostasis but subsequently change the focus of their study to investigate how internal state influences taste-evoked warm preference in hungry flies. Using mutants for NPF and sNPF the authors demonstrate that both peptides play a pivotal role in taste-evoked warm preference after sucrose feeding but not for nutrient-induced warm preference. Similarly, they found that DH44, AKH and dILP6, Upd2 and Upd3 neurons are also required for taste-evoked warm preference but not for nutrient-induced warm preference. Here again, the authors do not keep the systems stable and change between inhibition of neurons through Kir and mutants for peptides. For a better comparison, it would be preferable to use always exactly the same technique to inhibit neuron signalling.

It would be interesting to find the neural circuity of thermo-sensation and metabolic homeostasis, but we do not have any luck so far. We will continue to look into the neural circuits which control taste-evoked warm preference and nutrient-induced warm preference. Since UAS-Kir is such a strong reporter, it may kill the flies sometime. So we couldn't use UAS-Kir for all Gal4 flies.

DH44 is expressed in the brain and in the abdominal ganglion where they share the expression pattern with 4 Lk neurons per hemisphere. Seeing the impact of Lk signalling in metabolism (AlAnzi et al., 2010) the authors should provide evidence that the observed effect is indeed because of DH44 and not Lk.

It would be interesting to see if Lk may play a role in taste-evoked warm preference and/or nutrient-induced warm preference. We would like to systematically screen which neuropeptides and receptors are involved in the behavior in the next study.

Seeing the results on dILP6 it is interesting that Li and Gong (2015) could show in larvae that cold-sensing neurons directly interact with dILP neurons in the brain. It would be interesting to see whether similar circuitry may exist in adult flies to regulate temperature preferences and these peptidergic neurons. Further, it appears interesting that again these animals need much longer time to display the observed shift in temperature (which again should be clearly indicated in the figure legend too). These observations should be more carefully considered in the discussion part too.

We have revised the manuscript.

In the last part of the study, the authors investigate how sensory input from temperature-sensitive cells may transmit information to central clock neurons and how these in turn may influence temperature preference under starvation. The experiments assume that DH44-expressing neurons play a role in the output pathway of the central clock. Using the clock gene null mutants per and tim the authors show that even though the animals display a significant starvation response neither per nor tim mutants exhibited taste-evoked warm preference, indicating a taste but not nutrient-evoked temperature preference regulation.The authors demonstrate interesting new data on how taste input can influence temperature preference during starvation. They propose how gustatory pathways may work together with thermosensitive neurons, peptidergic neurons and finally try to bridge the gap between these neurons and clock genes. The study is very interesting and the data for each experiment alone are very convincing. However, in my opinion, the authors have opened many new questions but did not fully answer the initial question - how do taste-sensing neurons influence temperature preferences? What are the mechanisms underlying this observation? Instead of jumping from gustatory neurons to thermosensitive neurons to peptidergic neurons to clock genes, the authors should have stayed within the one question they were asking at the beginning. How does sugar sensing influence the physiology of thermos-sensation? Before addressing all the following questions of the manuscript the authors should first directly decipher the neuronal interplay between these two types of neurons.

Thank you for your suggestion. It would be interesting to find the neural circuity of thermo-sensation and metabolic homeostasis. We have tried but there is no luck so far.

The authors could e.g., employ Ca or cAMP-imaging in anterior or cold-sensitive cells and see how the responsiveness of these cells may be altered after sugar feeding. Or at least follow the idea of Li and Gong about the thermos-regulation of dILP-expressing neurons.

Thank you for your suggestion. Since we do not know how dlLP-expression neurons are involved in temperature response in the adult flies. We will focus on the cells using Calcium imaging for the next study.

Anatomical analysis using the GRASP technique may further help to understand the interplay of these neurons and give new insights into the circuitry underlying food preference alteration under starvation.

Thank you for your suggestion. It would be interesting to find the neural circuity of thermo-sensation and metabolic homeostasis. We have tried but there is no luck so far.

Minor comments:Line 51: Hungry animals are desperate for food - I think the authors should not anthropomorphize at this point too\ much but rather strictly describe how the animals change their behavior without any interpretation of the mental state of the animal.

We have modified the manuscript.

Line 80: Hunger and satiety dramatically affect animal behavior and physiology and control feeding - please not only cite the papers but also give a short overview of the cited papers on which behaviors are altered and how.

We have revised the manuscript.

Overall statistic: The authors do comparative statistics always against starved animals throughout but often state in the text a comparison against fed (Line 111: "but did not reach that of the fed flies") I think the authors should describe the date according to their statistics and keep this constant throughout the paper.

Sorry for the confusion. We originally had it, but we removed it. We have added the additional statistical analyses.

Figure legends: Overall the figure legends could be more developed and more detailed.

We have revised the manuscript.